# Use of amplicon-based sequencing for testing fetal identity and monogenic traits with Single Circulating Trophoblast (SCT) as one form of cell-based NIPT

Xinming Zhuo[1]*, Qun Wang[1], Liesbeth Vossaert[1], Roseen Salman[1], Adriel Kim[2], Ignatia Van den Veyver[1,3], Amy Breman[4], Arthur Beaudet[1]*

1 Department of Molecular and Human Genetics, Baylor College of Medicine, Houston, TX, United States of America, 2 Graduate Program in Diagnostic Genetics, MD Anderson Cancer Center, Houston, TX, United States of America, 3 Department of Obstetrics and Gynecology, Baylor College of Medicine, Houston, TX, United States of America, 4 Department of Medical and Molecular Genetics, Indiana University School of Medicine, Indianapolis, IN, United States of America

* abeaudet@bcm.edu (AB); xmzhuo@gmail.com (XZ)

## Abstract

A major challenge for cell-based non-invasive prenatal testing (NIPT) is to distinguish individual presumptive fetal cells from maternal cells in female pregnancies. We have sought a rapid, robust, versatile, and low-cost next-generation sequencing method to facilitate this process. Toward this goal, single isolated cells underwent whole genome amplification prior to genotyping. Multiple highly polymorphic genomic regions (including HLA-A and HLA-B) with 10–20 very informative single nucleotide polymorphisms (SNPs) within a 200 bp interval were amplified with a modified method based on other publications. To enhance the power of cell identification, approximately 40 Human Identification SNP (Applied Biosystems) test amplicons were also utilized. Using SNP results to compare to sex chromosome data from NGS as a reliable standard, the true positive rate for genotyping was 83.4%, true negative 6.6%, false positive 3.3%, and false negative 6.6%. These results would not be sufficient for clinical diagnosis, but they demonstrate the general validity of the approach and suggest that deeper genotyping of single cells could be completely reliable. A paternal DNA sample is not required using this method. The assay also successfully detected pathogenic variants causing Tay Sachs disease, cystic fibrosis, and hemoglobinopathies in single lymphoblastoid cells, and disease-causing variants in three cell-based NIPT cases. This method could be applicable for any monogenic diagnosis.

## Introduction

Since 2015, several influential professional societies, including the International Society for Prenatal Diagnosis (ISPD) and American College of Obstetricians and Gynecologists (ACOG), have stated that noninvasive prenatal testing (NIPT) is an available screening option for all pregnant women. Current NIPT is based on analysis of cell-free fetal (cff) DNA, and it has

**Data Availability Statement:** All relevant data are within the paper and its Supporting Information files.

**Funding:** This work was supported by internal institutional funds at Baylor College of Medicine awarded to AB and National Institutes of Health in the form of a grant awarded to IVdV (HD055651-12S1) at Baylor College of Medicine.

**Competing interests:** The Houston authors are or were faculty and staff at Baylor College of Medicine (BCM), which is a partial owner of a for profit diagnostic company, Baylor Genetics (BG); Houston authors are also employees of or have advisory or lab director roles at BG. ALB is founder and CEO of Luna Genetics, Inc., but Luna had no involvement in this work. This does not alter our adherence to PLOS ONE policies on sharing data and materials.

become widely available since its introduction to clinical practice in 2011. In contrast, cell-based NIPT, which relies on the isolation of circulating fetal cells in maternal blood, has been a long-sought alternative to cell-free NIPT and is now approaching commercialization. Currently, the cell-free NIPT approach has the advantage of a faster turnaround time and lower cost. However, the accuracy of cell-free NIPT is impacted by the large amount of maternal DNA in plasma (more than 80% of all circulating DNA) and the highly fragmented nature of this genetic material. Thus, it is only recommended for detection of the common fetal aneuploidies by many professional societies [1, 2]. These drawbacks can be addressed by cell-based NIPT, but it is not yet available as a clinical test. Cell-free NIPT, trophoblast-based NIPT and CVS all can detect placental mosaicism, while amniocentesis, fetal nucleated red blood cell (fnRBC)-based NIPT, fetal blood sampling, and amniocentesis can help to clarify whether mosaicism involves the fetus or is confined to the placenta. Cell-based NIPT would potentially have a higher positive predictive value compared to cell-free NIPT, since the DNA source is purely fetal or placental in origin without any maternal contamination [3–5]. Limitations of cost and throughput would need to be overcome for cell-free NIPT to be a routine alternative. Recently, multiple groups have reported successful cases of cell-based NIPT via capturing trophoblast cells [3, 6, 7].

The critical step for cell-based NIPT is the recovery of rare fetal cells, such as trophoblasts. As described previously [3, 4], 30–40 mL of blood is collected at 10–16 weeks' gestation, followed by density fractionation or magnetic activated cell sorting (MACS) with anti-trophoblast antibodies to enrich the nucleated cells. Then, the nucleated cells are immunostained to identify trophoblasts that are cytokeratin positive and leukocyte common antigen (CD45) negative. The stained cells are picked individually under fluorescence microscopy with an automatic instrument described previously [3, 4] and subjected to whole genome amplification (WGA), which allows downstream genotyping, and copy number analysis using array Comparative Genomic Hybridization (CGH) or next generation sequencing (NGS).

Genotyping is an essential step after isolating the putative fetal cells. Typically, a successful cell-based NIPT would isolate 5–10 cells per 30 mL maternal blood sample. Since the nucleated cell recovery is a complicated multiple-step procedure, and several antibodies are used, there is a chance of picking a maternal cell (~10% in our experience). Whole genome shotgun (WGS) sequencing at low coverage (5–10 million reads per cell) provides good copy number data, but it does not readily distinguish fetal and maternal cells if the fetus is female. Previously, we used short tandem repeat analysis, SNP arrays, or Y-chromosome targeted qPCR to confirm the fetal origin of single cells. However, there are various disadvantages to these approaches, such as inefficiency, ambiguity, high cost, or limited application.

In this work, we developed a fast, low-cost, and reliable genotyping assay with amplicon sequencing. We sequenced approximately 90 highly polymorphic SNPs within about 40 amplicons. Among these amplicons, four contain multiple common SNPs (see Materials and Methods), which allow for haplotyping of the WGA DNA product. Together, it allows for the effective differentiation of cells containing the fetal genome from cells of maternal origin in most cases. This genotyping uses a small aliquot of the WGA product and does not interfere with downstream analysis. This method could also be easily expanded for the detection of additional disease-associated variants, which would have clinical utility for pregnancies with increased risk for monogenic disorders.

## Materials and methods

### Sample collection and preparation

Blood samples were collected from pregnant women from multiple centers under a protocol approved by the Baylor College of Medicine or Columbia University Medical Center

Institutional Review Boards utilizing written informed consent. Approximately 30 mL of blood was collected into anticoagulant EDTA Vacutainer tubes (BD). Fetal cells were enriched with methods described in Breman et al., 2016. Both cytokeratin (CK)-positive putative fetal cells, and CK-negative maternal white blood cells were picked from maternal blood using the CytePicker® equipment (Rarecyte). There were 154 usable blood samples from 152 pregnancies; two women had two samples collected during one pregnancy.

## Overall strategy

The typical cell-based NIPT workflow yields 3–10 singlet or doublet cells per patient, individually captured in a PCR tube for downstream WGA. The WGA DNA products of those cells must be checked for quality and confirmed as nonmaternal cells before finalizing the interpretation. Thus, a fast, low-cost, and high-throughput genotyping assay is necessary. To meet this need, we designed a single-cell genotyping assay using a modified amplicon sequencing approach for genotyping (Fig 1). The first step is conventional PCR of a pool of amplicons with bridging adaptors, which contains a partial sequence of Illumina i5 and i7 adaptors. The second step is adding a dual index with a sequencing adaptor to previous PCR products. This concludes the library construction for the Illumina machine. DNA samples with different indexes were balanced and pooled for sequencing with Illumina Miseq. The sequencing result was demultiplexed with Illumina BaseSpace. The demultiplexed reads were mapped with conventional BWA-MEM. The mapped reads were used for SNP typing and amplicon haplotyping, the details of which are discussed below.

## Amplicon design

Three groups of amplicons were used to carry out amplicon-seq. The first group consisted of amplicons with multiple common SNPs (>5% prevalence) as suggested in Debeljak et al. [8], including regions in HLA-A, HLA-B, chromosome 7q11 and chromosome 11q22, which have good sequencing coverage in single-cell WGA; all of these amplicons have at least eight common SNPs. The information from these common SNPs can be used for effective haplotyping and identifying the origin of isolated cells. The second group, which consisted of 37 amplicons, contained common SNPs selected from the Human Identification panel (ABI), which covers

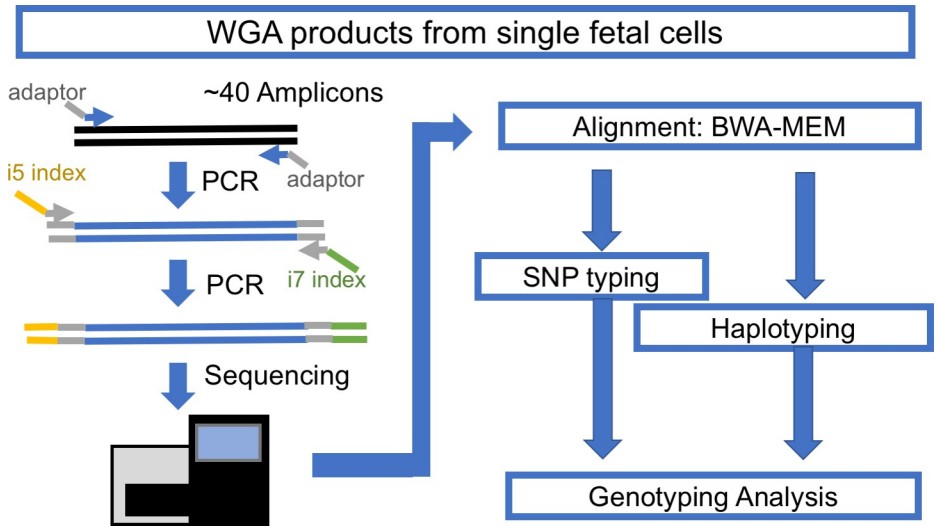

**Fig 1. The workflow for amplicon-based genotyping.**

most chromosomes, including the Y chromosome. They were selected according to sequencing coverage in the single-cell WGA product, which has a high tendency for dropout. The third group consisted of amplicons designed to detect single nucleotide variants associated with certain inherited disease genes of interest, including hemoglobin subunit beta (*HBB*), hexosaminidase A (*HEXA*), and cystic fibrosis transmembrane conductance regulator (*CFTR*) (Table 1). For studies of trophoblasts from specific at-risk cases, we also prepared amplicons for *DHCR7* and *RASPN* (Table 1). The primers of these amplicons were prepared with adaptors compatible with Illumina True-seq HT i5 and i7 adaptors.

## Library construction for Illumina

WGA was performed using the PicoPLEX kits (Rubicon/Takara) or, in some cases, the Ampli1 WGA kit (Silicon Biosystems), according to the manufacturer's protocol. Briefly, 100ng of WGA DNA product was used for Amplicon-seq library construction, with two-step PCR. The first step included 20–25 cycles of PCR with NEB Q5HS to amplify amplicons of interest and add designated adaptors. The second step used a previous adaptor sequence as primer binding sites to add Illumina i5 and i7 dual-index adaptors with 10–15 cycles of PCR. The 2-step PCR products were purified with standard AMPure protocol (Beckman) for the 100–300 bp product. PCR products were visualized by gel electrophoresis followed by the Bioanalyzer (Agilent) to check the quality and then quantified by using a KAPA Library Quantification Kit (Kapa Biosystems).

## Sequencing with Miseq

The barcoded DNA library was diluted to 2 nM and pooled together for denaturing with the Illumina protocol. An 8–10 pM diluted denatured library was mixed with 5–10% PhiX control. The mixed library was loaded onto the Illumina Miseq with 150-cycle v3 kits (Illumina) and sequenced with 2x76 reads and dual index.

## Demultiplexing, alignment and variant calling

Sequencing results were demultiplexed by Illumina BaseSpace. The reads were mapped and processed with a shell script (S3 File). In summary, the fastq.gz raw files were aligned with customized reference files (S1 File) by BWA-MEM [9]. Samtools [10, 11] and Bam-readcount followed by a customized R script (S4 File) were used for calling variants within selected intervals (S2 File). The cutoff for calling a variant is at least ten reads of the less frequent allele and 5% of all reads.

Variants from each sample were summarized and compared with their paired control with an R script (S6 File). Typically, a cell with at least 2 SNPs and at least 6% of comparable SNPs different from its maternal gDNA control is scored as a fetal cell. Otherwise, it will be classified as an uninformative cell. Throughout this manuscript, a cell or a SNP is referred to as informative if the putative fetal cell has an allele not carried by the mother (e.g., mother is AA and the putative fetal cell is AB or _B). An uninformative cell does not have alleles not carried by the mother and may be a maternal cell or a fetal cell with inadequate genotyping data. This may represent an underestimate of fetal cells, especially when one but not two SNPS support fetal origin.

## Haplotype calling

Using the Illumina 2x76 read length necessitated performing stitch overlap for read 1 and read 2 with PEAR (Paired-end read merger) [12] for amplicons shorter than 150 bp. For amplicons

**Table 1. List of amplicons.**

| Amplicon_ID | Chr | Amplicon_Start | Amplicon_Stop | Size | Annotation |
|---|---|---|---|---|---|
| rs1490413 | chr1 | 4367241 | 4367415 | 175 | |
| rs4847034 | chr1 | 105717572 | 105717689 | 118 | |
| rs3780962 | chr10 | 17193291 | 17193386 | 96 | |
| rs964681 | chr10 | 132698373 | 132698467 | 95 | |
| rs1498553 | chr11 | 5708942 | 5709103 | 162 | |
| rs901398 | chr11 | 11096160 | 11096278 | 119 | |
| rs2269355 | chr12 | 6945833 | 6946005 | 173 | |
| rs4530059 | chr14 | 104769098 | 104769197 | 100 | |
| rs2016276 | chr15 | 24571747 | 24571845 | 99 | |
| rs2342747 | chr16 | 5868655 | 5868769 | 115 | |
| rs2292972 | chr17 | 80765753 | 80765827 | 75 | |
| rs938283 | chr17 | 77468418 | 77468592 | 175 | |
| rs9905977 | chr17 | 2919336 | 2919452 | 117 | |
| rs1024116 | chr18 | 75432299 | 75432467 | 169 | |
| rs576261 | chr19 | 39559774 | 39559848 | 75 | |
| rs12997453 | chr2 | 182413125 | 182413299 | 175 | |
| rs1005533 | chr20 | 39487029 | 39487201 | 173 | |
| rs445251 | chr20 | 15124851 | 15125009 | 159 | |
| rs221956 | chr21 | 43606946 | 43607048 | 103 | |
| rs2830795 | chr21 | 28608067 | 28608212 | 146 | |
| rs733164 | chr22 | 27816739 | 27816833 | 95 | |
| rs1355366 | chr3 | 190806053 | 190806219 | 167 | |
| rs4364205 | chr3 | 32417580 | 32417720 | 141 | |
| rs6444724 | chr3 | 193207331 | 193207425 | 95 | |
| rs1979255 | chr4 | 190318032 | 190318131 | 100 | |
| rs159606 | chr5 | 17374818 | 17374977 | 160 | |
| rs338882 | chr5 | 178690682 | 178690774 | 93 | |
| rs7704770 | chr5 | 159487871 | 159488033 | 163 | |
| rs13218440 | chr6 | 12059906 | 12060004 | 99 | |
| rs6955448 | chr7 | 4310289 | 4310423 | 135 | |
| rs1360288 | chr9 | 128967996 | 128968115 | 120 | |
| rs1463729 | chr9 | 126881396 | 126881493 | 98 | |
| rs7041158 | chr9 | 27985851 | 27986020 | 170 | |
| P256 | chrY | 8685171 | 8685289 | 119 | |
| rs17250845 | chrY | 8418867 | 8418960 | 94 | |
| rs35284970 | chrY | 2734829 | 2734921 | 93 | |
| rs3911 | chrY | 21733328 | 21733502 | 175 | |
| HLA-A | chr6 | 2911156 | 2911140 | 245 | Haplotype |
| HLA-B | chr6 | 31319491 | 31319646 | 156 | Haplotype |
| Chr7q11 | chr7 | 64895160 | 64895374 | 215 | Haplotype |
| Chr11q22 | chr11 | 99491336 | 99491527 | 192 | Haplotype |
| rs334&rs33930165 | chr11 | 5226980 | 5227053 | 74 | *HBB* |
| rs11393960 | chr7 | 117559481 | 117559645 | 165 | *CFTR* |
| rs75527207&rs74597325 | chr7 | 117587771 | 117587870 | 100 | *CFTR* |
| rs121907954 | chr15 | 72350490 | 72350617 | 128 | *HEXA* |
| rs387906309 | chr15 | 72346528 | 72346691 | 164 | *HEXA* |
| rs147324677 | chr15 | 72346182 | 72346293 | 112 | *HEXA* |

*(Continued)*

**Table 1.** (Continued)

| Amplicon_ID | Chr | Amplicon_Start | Amplicon_Stop | Size | Annotation |
|---|---|---|---|---|---|
| **rs138659167** | Chr11 | 71146795 | 71146921 | 127 | *DHCR7* |
| **rs104894299&rs761584017** | Chr11 | 47469333 | 47469720 | 388 | *RASPN* |

The coordinates in this table include the first and last base of the amplicon so that subtraction of one number from the other gives a number one less than the product size.

larger than 150 bp, we stitched non-overlapping reads with a 15-N padding sequence. Those stitched reads were mapped with BWA-MEM with modified parameters that allow a bigger unmatched gap inside a mapped read. The mapped results were processed with an R script (S5 File) used in the main shell script to extract SNPs and reconstitute a new sequence with CIGAR information. The new concise sequences were tabulated and grouped with sequence similarity according to their Levenshtein distance. We assigned a haplotype to each major group of a concise sequence. Pair-wise haplotype comparison between the maternal gDNA sample and the putative fetal cells were performed with another R script (S7 File). All scripts are hosted and maintained on https://github.com/xmzhuo/NIPT_genotyping.

## Results

### Target region coverage

We compared the coverage of our amplicons for SNPs with gDNA and NIPT cell WGA products (Fig 2). For gDNA, most of the samples have very high coverage, which reflects the distribution of samples with many scorable SNPs. The WGA product of maternal white blood cells (WBCs) shows less scorable SNPs than gDNA, which would be the result of starting with a single diploid genome target in combination with fixation, staining, and amplification during WGA.

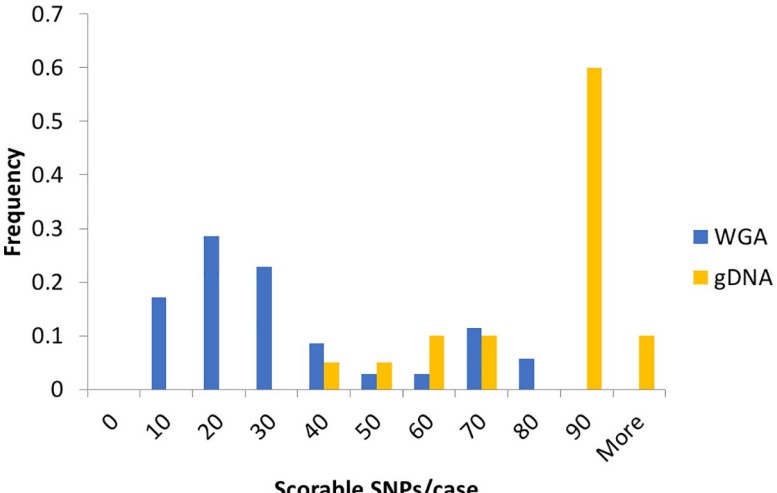

**Fig 2. Comparison of the coverage of several genomic DNAs and NIPT single cells.** The count of samples with various scorable SNPs was normalized to the total number of samples of each group (gDNA n = 20, WGA n = 35). The scorable SNP cutoff was at least 5% minor allele frequency (MAF) and ten reads. Data include NIPT case numbers 946, 977, 982, 983, 984, 988, 989, 990, 991, 992, 993, 996, and 998 (clinical data in S8 File).

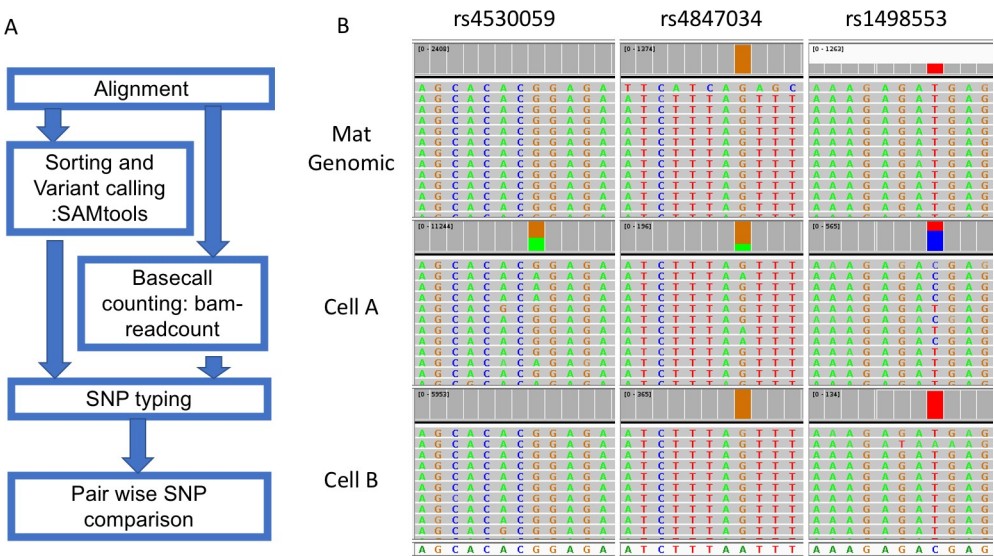

**Fig 3. The workflow and an example of SNP typing.** Reads for the variant allele are colored green while reads for the normal allele are colored grey and summed in red. The mother is homozygous for all three SNPs, while cell A is heterozygous for all three SNPs in each case having an allele that the mother does not have. Cell B is homozygous for all SNPs indistinguishable from the mother and is interpreted as maternal or noninformative (fetal with allele drop out for all three SNPs). Data from NIPT case number 1000 (clinical data in S8 File).

## SNP typing of NIPT WGA products

A typical process of SNP typing after BWA-MEM aligner mapping includes variant calling with Samtools and retrieving the read depth with bam-readcount (Fig 3). This step will produce a table containing variant call information including allele fraction and read depth of all SNPs of interest. Information on indels was masked to avoid confusion in later steps. Then, we performed the pair-wise comparison of WGA products with the maternal gDNA. The script will calculate how many SNPs are different between two DNA samples. In an example case with two cells (Fig 3), three SNPs show a difference between one of the cells (middle panel) and maternal gDNA, which suggests that it is likely to be a fetal cell. The other cell shows identical calls with maternal gDNA, which suggests that it either is a maternal cell or a noninformative fetal cell.

The results for 152 cases can be divided into three groups as shown in Table 2. First, there were 7/152 (4.6%) cases where none of the cells passing WGA could be proven to be fetal; Second, there were 42/152 (27.6%) cases with only one cell scored as fetal; and third, there were 103/152 (67.8%) with two or more cells scored as fetal as defined in methods. At the single cell level for cells that pass WGA, cells can be scored in only two ways: 1) uninformative meaning no or inadequate evidence of fetal status or 2) adequate evidence for fetal status. Some of the "uninformative" cells were certainly maternal, and we cannot distinguish an uninformative

**Table 2. Tabulation of cases according to outcome of genotyping.**

| Condition | 2 SNP + 10% | 2 SNP + 8% | 2 SNP + 6% |
|---|---|---|---|
| None of the available cells with adequate WGA were proven fetal. | 13 (8.6) | 8 (5.3%) | 7 (4.6%) |
| Only one fetal cell identified. | 46 (30.3%) | 44 (28.9%) | 42 (27.6) |
| Two or more fetal cells identified. | 93 (61.2%) | 100 (65.8%) | 103 (67.8%) |
| Total cases/pregnancies | 152 | | |

fetal cell from a maternal cell. Deeper genotyping would make healthy uninformative fetal cells informative, but some fetal cells are known to be apoptotic with degraded DNA and may or may not pass WGA. False positives and false negatives at a single cell level are discussed below. This method is a work in progress, and we consider one fetal cell as partial success and two or more fetal cells as success from a clinical perspective. Improved cell recovery and improved genotyping would be needed to achieve an optimal test. The relative roles of failure to isolate and amplify cells, uninformative genotyping, and suboptimal numbers of fetal cells can be calculated from Table 2.

In Table 2, we examined what percent of cases had one or more or two or more cells scored as fetal. Individual cells were scored as fetal if two or more SNPs had at least 10% reads for an allele that was not present in the mother. Cases were then subdivided into those where the informative SNPs indicating fetal status were at least 10%, 8%, or 6%, of the scorable SNPs (There were no additional cases where the informative SNPs were less than 6% of the scorable SNPs). There were 60 cells with one informative SNP suggesting that requiring two SNPs may undercount fetal cells. It is important to distinguish the percent of cases (82.4%) that had two or more fetal cells (103/152) from the percent of cells (78.7%) that had two or more fetal SNPs indicating fetal status (408/518).

If one assumes that the NGS data are 100% reliable for sex based on X and Y data, which we believe is the case, the sensitivity and specificity of the genotyping can be estimated from normal male singleton pregnancies as shown in Table 3. Definition of other contingencies are included in the same table. Identifying a fetal cell as fetal by genotyping and confirming that it is male based on NGS is a true positive. The results vary depending on the cutoff for scoring a SNP allele as present. If we accept two different 2 SNP + 6% cutoff, based on normal male singleton pregnancies the true positive rate was 82.1%, true negative 6.6%, false positive 3.3%, and false negative 7.9%. This is equal to a sensitivity [True Positive Rate = TP/(TP+FN) of 91.2% (124/124+12)] and a precision [Positive Predictive Value = TP/(TP+FP) of 96.1% (124/124 +5)]. These results would not be sufficient for clinical diagnosis, but they demonstrate the general validity of the approach and suggest that deeper genotyping of single cell could be completely reliable.

In NIPT #1000 (Fig 4), we isolated eight putative fetal cells from the mother's blood for WGA. Cells G78, G212, G 227, and G232 all have at least five SNPs which differed from the maternal gDNA; thus, they were confirmed to be fetal in origin. The remaining three cells (G79, G113 and G320) had 0–1 SNPs which differed from the maternal gDNA, and these were considered uninformative and possibly white blood cells accidently isolated from the mother's blood. Cell G106 has only 2 SNP difference in less than 20 informative SNPs, which was considered as a low-quality sample and uninformative.

**Table 3. Sensitivity and specificity for genotyping**

| | Normal male singleton | | | |
|---|---|---|---|---|
| | True positive | True negative | False positive | False negative |
| | Cells scored as fetal by genotype and male by NGS | Cells scored as not fetal by genotype and female by NGS | Cells scored as fetal by genotype but female (maternal) by NGS | Cells scored as not fetal by genotype but male by NGS |
| 2 SNP | 126 | 10 | 5 | 10 |
| 2 SNP + 6% | 124 | 10 | 5 | 12 |
| 2 SNP + 8% | 117 | 10 | 5 | 19 |
| 2 SNP + 10% | 107 | 10 | 5 | 29 |

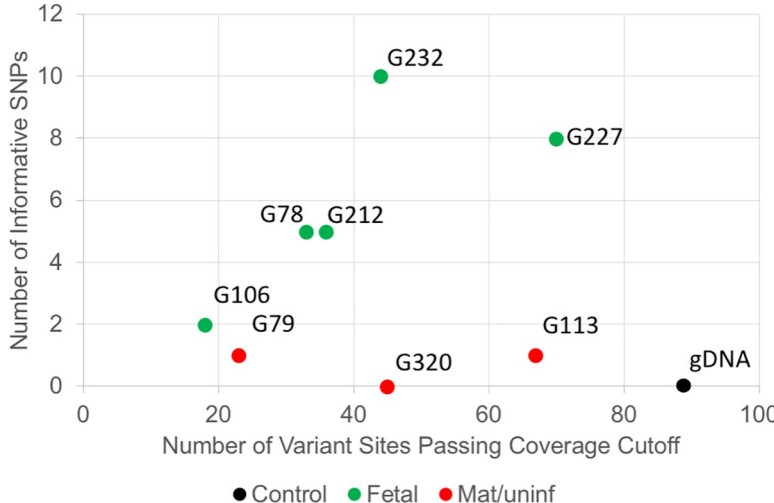

**Fig 4. Example of SNP typing result of one NIPT case.** The X-axis indicates the number of variant sites passing coverage cutoff. The Y-axis is the number of informative SNPs in a cell. Data from NIPT case number 1000.

## Haplotyping of NIPT WGA products

The haplotyping of NIPT WGA products potentially has higher power than SNP typing for identifying fetal cells. Since the WGA product typically went through the 14–16 cycles of amplification from trace amounts of input DNA, there is a small chance of introducing new mutations, which would affect the precision of SNP typing at low read-depth. For example, some of the cells from case #1000 (Fig 4) had one low-depth SNP difference from the maternal gDNA. To address this issue, we developed a haplotyping approach for multiple highly polymorphic regions, which contain multiple very common SNPs within the 200 bp amplicon. Thus, we can decrease the impact of random mutations introduced during the WGA process on the final interpretation (i.e., one nucleotide change is less likely to change the classification of a major haplotype group, which is comparable to HLA typing approach). Mutations introducing a random change are not rare, but mutations switching from one allele at a SNP to the other polymorphic allele are much rarer. In addition, the haplotypes of an amplicon can be treated as a permutation of a given number of SNPs, which theoretically generates much more haplotypes than SNP types and has higher power at differentiating two cells. Third, we can estimate the point at which a sequence artifact arose based on the fraction of each minor haplotype group in the total reads for a given amplicon. For example, a high fraction indicates a variant preexisted in the cell, a medium fraction indicates the variant arose during the WGA step, while a low fraction is consistent with an artifact from the final step of amplicon-seq.

We performed haplotyping with the following steps (Fig 5). After regular alignment with BWA-MEM with the default setting, we joined the Read 1 and Read 2 with PEAR [12]. The overlapping Read 1 and Read 2 were merged. If the amplicon is longer than two reads joined together, we merged the two reads and padded the gap with a tandem repeat of N. The merged reads were remapped with BWA-MEM again with a lenient setting to tolerate a larger gap. The remapped reads were processed with an R script to extract the selected SNPs, and each read was reconstituted with the concise sequence while preserving the read ID. The concise reads were then tallied and ranked according to frequency (typically, only the top 10 were kept, which usually consist of more than 99.99% of all types of reads). The Levenshtein distances were calculated for these reads, which typically ended up with only one or two major

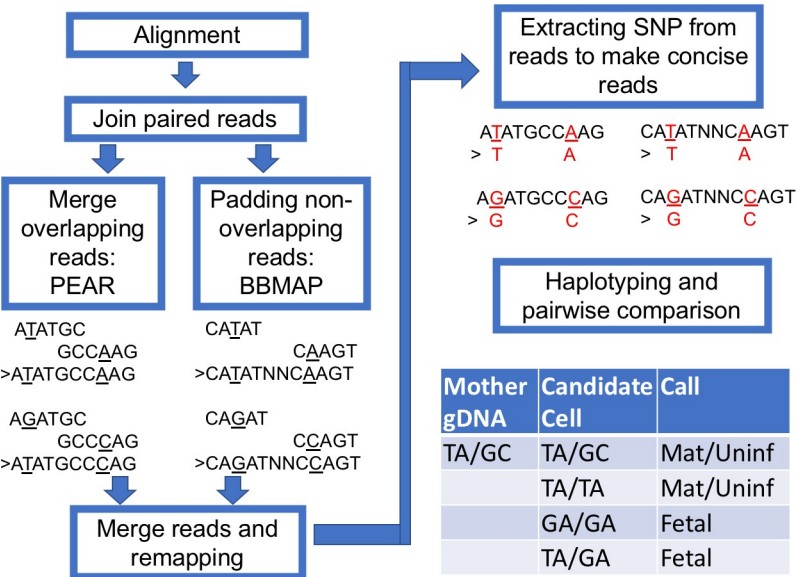

**Fig 5. The workflow for haplotyping.** The paired aligned reads were jointed with PEAR and padded with BBMap short read aligner when a gap exists. The new joint reads were remapped with BWA-MEM with a modified configuration. The highly polymorphic sites were extracted for constructing concise haplotypes. To demonstrate the workflow, we present example short reads with TA and GC haplotypes.

groups to represent the haplotype of this amplicon. To compare the haplotypes of more than two samples, all the top reads of each sample were pooled together, and their distances were calculated, which will determine if these samples share the same read group (haplotype). For example, the maternal gDNA carried a mocked haplotype 1 (TA) and haplotype 2 (GC). A positive fetal cell should be identified to carry at least one new haplotype 3 (GA), which would be TA/GA or GA/GA (result from dropout of haplotype 1 or 2) (Fig 5).

The haplotyping approach can effectively differentiate a candidate cell from maternal gDNA. In the case shown in Fig 6, we have the gDNA from both parents. As described previously, we extracted all 28 SNP sites in the HLA-A amplicon and reconstructed a concise 28 nt sequence for each read. In this case, the top four most frequent read types of potential fetal cells can be grouped into two major groups, with a Levenshtein distance of more than 2 (S1 Fig). The intra-group difference has a distance of less than 0.5, which suggests a difference of only one nucleotide. The difference likely results from artifacts introduced during extensive amplification (WGA then PCR). The same condition was observed in maternal gDNA and paternal gDNA as well. We observed the inheritance pattern of haplotypes when all read types from maternal, paternal, and fetal DNA were plotted together. One fetal haplotype matched with the mother and the other matched with the father. From these haplotype groupings, we concluded that this is a true fetal cell.

We tested the performance of haplotyping in four amplicons with matched gDNA, WBC, and fetal cells (S2 Fig). For gDNA, all four amplicons performed nicely to distinguish one from the other. For WBC and fetal cells, the performance was not as good, largely owing to the dropout events. However, when four amplicons were combined, we can still distinguish about 50% of all the cells (from selected cases with both fetal cells and WBCs) (S3A Fig). With combined power of both SNP typing and haplotyping, we can increase the solving rate of differentiating a WBC from around 60% to more than 70% (from cases independent of fetal cell existence) (S3B Fig).

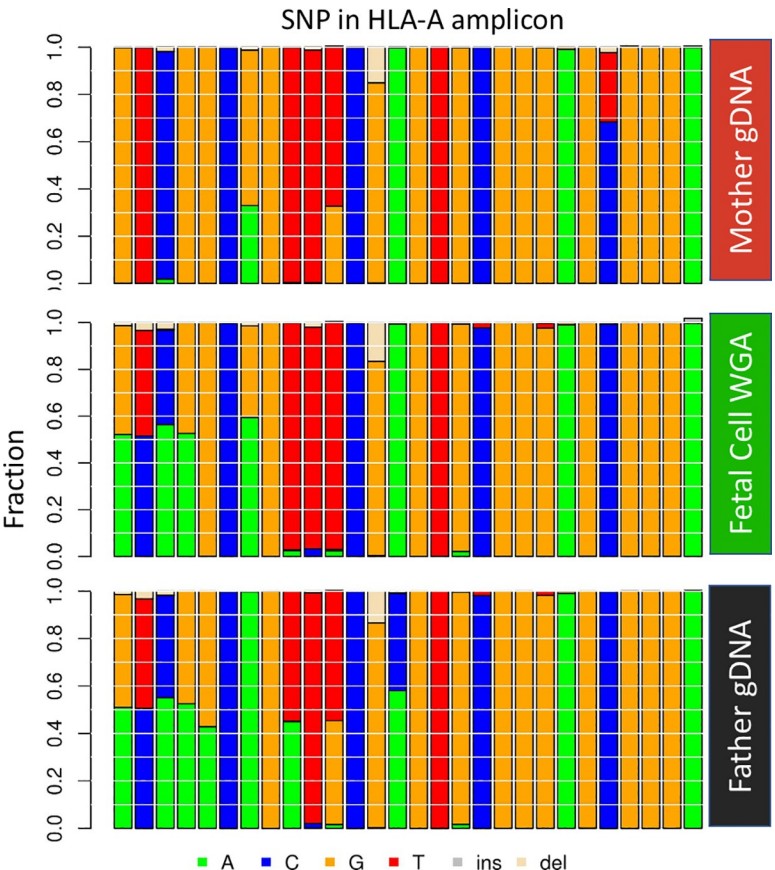

**Fig 6. Haplotype analysis of a fetal cell compared to both parents.** Each vertical column represents a particular SNP in the haplotype. The left-most four SNPs are informative as the fetal cell has an allele that the mother does not have and indicate that the putative fetal cell is indeed fetal. The 24 SNPs to the right are not informative for the fetal cell as they do not have an allele that the mother does not have. Data from NIPT case number 1000.

## Genotyping for monogenic disease-causing variants

We also wished to use this method to genotype for monogenic disease-causing variants. We first evaluated the ability to detect disease variants in single cultured lymphoblasts of known genotypes. Our cell based NIPT provides pure fetal DNA, which allows us to look at mono-genic disease-causing variants. Here, we developed amplicons that contain *HEXA* c.805C>T, *HBB* c.19T>A, *HBB* c.20C>T, and *CFTR* c.1521_1523delTCT. Corresponding cells carrying certain variants were obtained from the Coriell Institute and single lymphoblasts were picked from tissue culture and processed for WGA. The cells were isolated and genotyped by methods described herein. We successfully detected the known variants in the WGA products of cells carrying these changes (Fig 7). The allele dropout rate was 15% and 8% for unfixed and fixed lymphoblasts, respectively. The higher rate of allele drop out in fetal cells is presumably caused by some combination of DNA degradation caused by time in the maternal circulation includ-ing apoptosis and by cell isolation including fixation and permeabilization steps.

In all of our cell-based NIPT samples studies, there were three families with known patho-genic variants. One was a family where the mother was affected with sickle cell anemia, and the fetus was expected to be heterozygous based on parental information. No amniocentesis or CVS was performed. In a second family, both parents were carriers with the same known path-ogenic variant in *DHCR7*. By amniocentesis, the fetus was heterozygous for the variant carried

**Fig 7. Testing for sequence variants in single lymphoblasts.** From left to right, Tay-Sachs, *HEXA* c.805C>T het; *HBB* c.19T>A and c.20C>T compound het; *CFTR* c.1521_1523delTCT het. The reads are shown in the IGV browser.

by both parents. A third family had a previous affected child with congenital myasthenic syndrome type 11 caused by biallelic, compound heterozygous pathogenic variants in *RAPSN*. By CVS, the fetus carried the paternal but not the maternal pathogenic variant. For the sickle cell family, nine fetal cells were recovered and four were genotyped. Two cells, G54 (Fig 8) and G532 (Alternative Allele Frequency (AAF) 17%, data not shown), were heterozygous for the pathogenic variant (Fig 8), while one cell (G474) had dropout for the normal allele and one cell (G309) did not pass coverage cutoff (data not shown). The cell-based NIPT data scored the fetus as heterozygous. For the *DHCR7* family, seven fetal cells were recovered and four were genotyped. One cell, (G1286), was heterozygous for the pathogenic variant (AAF 80%) (Fig 8), while two cells (G123 and G4584) had dropout for the normal allele and one cell (G1074) had dropout for the variant allele. Again, the cell-based NIPT data scored the fetus as heterozygous. For the RAPSN family, four fetal cells were recovered and four were genotyped. In Fig 8C, the pathogenic variant in each parent is shown. The G540 cell in Fig 8C, shows absence of the maternal pathogenic variant but presence of the paternal pathogenic variant. All four cells showed absence of the maternal pathogenic variant. For the paternal pathogenic variant, two cells (G540 and G2847 (AAF 87%)) were heterozygous while one cell (G1517) had dropout for the normal allele and one cell (G360) had dropout for the paternal pathogenic variant. There is a small probability that the fetus carries the maternal pathogenic variant, but there was dropout in all four cells; more likely the fetus does not carry the maternal pathogenic variant in agreement with the CVS data.

## Discussion

This single cell genotyping assay can provide an essential step for confirming the fetal origin of cells obtained from cell-based NIPT workflows. Through genotyping, we can reject cells that are of indeterminate or maternal origin and provide metrics for WGA DNA quality using multiple amplicons. Thus, researchers can focus on smaller numbers of cells, which can then be used for a more expensive downstream test or analysis, such as the low-depth NGS and microarray for CNV analysis [3, 4, 6]. We can add more amplicons to cover variants of interest for detecting recessive or dominant inherited diseases. The data demonstrate the principle that detailed genotyping of individual cells can distinguish fetal from maternal cells, but the data are limited, and new methods to more extensively compare the genotype of individual cells to the genotype of the mother are needed.

Although there is evidence that some cells from previous pregnancies can persist for decades, especially CD34+ cells [13], there is no evidence that trophoblasts can persist from previous pregnancies, and we expect based on the biology of these cells that they are unlikely to persist. A future method that provides deep genotyping of individual cells could distinguish same sex nonidentical twins, but the method described here would not be sufficient to accomplish such distinction reliably.

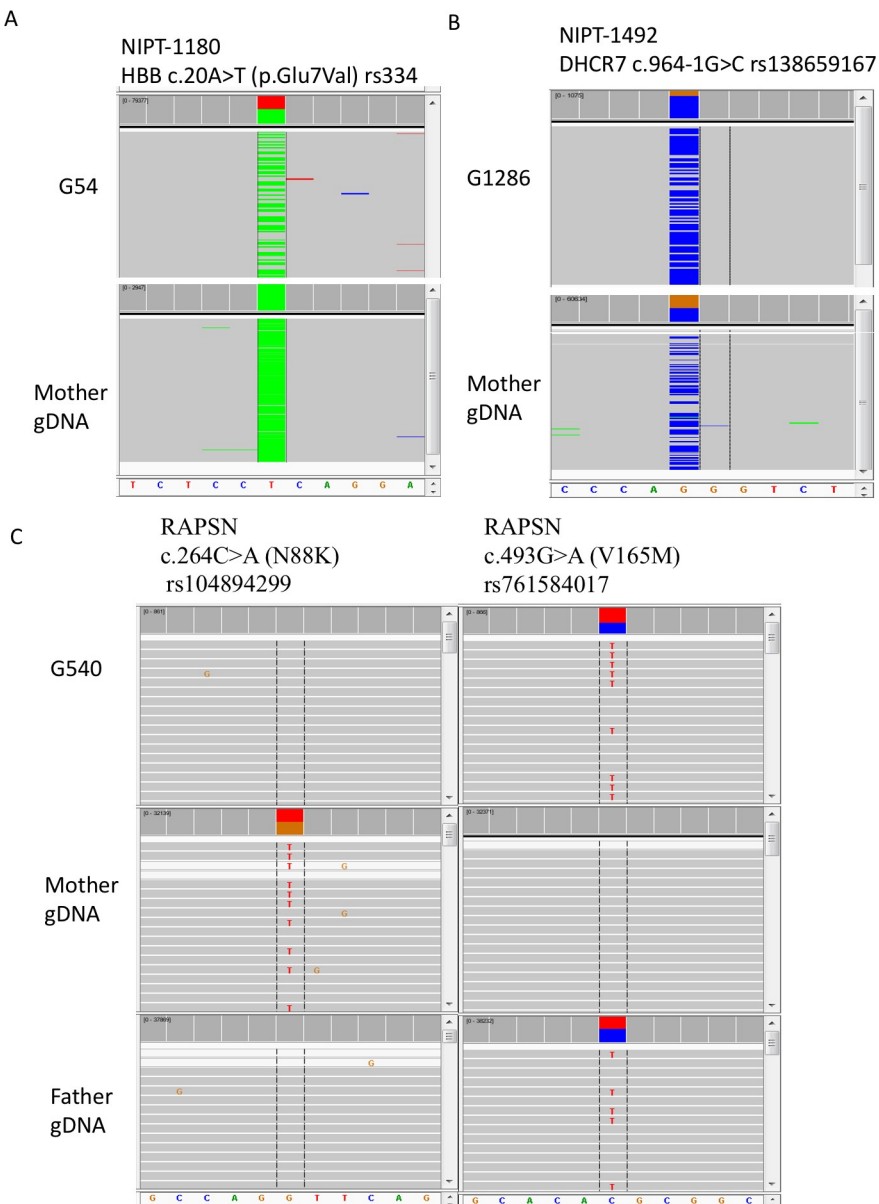

**Fig 8. Genotyping for pathogenic variants in trophoblasts from three cases.** In panel A, the mother is affected and homozygous for the sickle cell anemia variant. Fetal trophoblast G54 is heterozygous for the variant (ClinGen Accession: CA125138). Reads for the mutant allele are colored green while reads for the normal allele are colored grey and summed in red. In panel B, the mother is heterozygous for a *DHCR7* pathogenic variant that is also present in the father (CA090917). Fetal trophoblast G1286 is also heterozygous for the pathogenic variant, although there is biased over-representation of the variant allele. Reads for the mutant allele are colored blue while reads for the normal allele are colored grey and summed in brown. In panel C, the mother is heterozygous for the pathogenic N88K variant (CA199511) in the *RAPSN* gene, and the father is heterozygous for the V165M pathogenic variant (CA5976731). Fetal trophoblast G540 is heterozygous for the paternal V165K pathogenic variant but not for the maternal N88K pathogenic variant. Allele drop out for the N88K variant cannot be ruled out, and multiple cells must be tested to gain statistical evidence that the fetus has not inherited the N88K variant. All results agreed with data from amniocentesis or CVS. Data from NIPT case numbers 1180, 1492, and 1607.

Although this assay is promising, there are still some limitations. First, the dropout rate for individual amplicons is significant, which hinders the power of cell identification and affects the detection of disease-causing variants. Second, although the cost is relatively low and the

running time is short (<24 h), it still takes extra effort to complete and could increase the turn-around time of NIPT. Third, the multiple steps of PCR after WGA are prone to introduce errors that may cause ambiguity at SNP typing, although errors switching from one SNP allele to another or from a mutant to wild-type genotype or vice versa are very rare.

There are multiple options for reducing the dropout rate for the improvement of this assay. First, we can increase the size of the panel, which allows more amplicons to compensate for the dropout. Second, we can modify the amplicons to reduce the size of the amplicons or otherwise improve the amplification. Third, improved versions of WGA are being developed which can reduce allele dropout.

Use of this method for genotyping fetuses at risk for specific pathogenic variants is potentially feasible. Although cell-free NIPT is relatively straight forward for genotyping paternal pathogenic variant, it is more complex for determining the maternal contribution to the genotype, although this can be accomplished with more complex analysis [14–16]. Cell-free NIPT has been used to screen for *de novo* variants in a panel of genes [17]. Confirmation for *de novo* variants can be performed by specific reanalysis of maternal plasma if the mother is not mosaic for the variant. Single cell analysis of circulating trophoblasts can be used to determine the genotype of fetuses at risk for monogenic disorders, although allele dropout must be ruled out by analysis of multiple cells or it can be addressed using karyomapping as has been used in single cell preimplantation testing [18]. Hopefully improved methods for recovering fetal cells from mother's blood will reduce any failure due to lack of cells, but use of haplotyping with cell-free NIPT in combination with haplotype analysis or digital PCR, amniocentesis, and CVS would be three alternative strategies if cell-based NIPT fails.

Our genotyping assay has the potential to be used in many clinical research applications. As described previously, it can be used to identify a fetal cell in cell-based NIPT and screen for known disease-causing variants. It also has the potential to distinguish between same sex dizygotic twins. Furthermore, this technique would also be adapted for other single cell applications, such as circulating tumor cell analysis. A future improved version may hopefully reduce the dropout rate in amplicons and increase the coverage at regions of interest.

## Supporting information

**S1 Fig. Grouping multiple amplicon haplotypes for a family trio.** We use HLA-A amplicon haplotypes from samples present in Fig 6 to demonstrate how to identify the fetal cell. Haplotype groups of the mother (Red), fetal cell (Green), and father (Black). The Y-axis of bar graphs indicates the factions of total reads in each DNA types in different read groups. The tree cluster suggests the distance between read-groups according to Levenshtein distance calculation.
(TIF)

**S2 Fig. Using ROC-AUC to estimate the performance of haplotyping with various distance setting.** The Y-axis is the ROC-AUC distance to diagonal line from 0 to 1. The X-axis is the distance used for haplotype grouping. Three types of DNA were used for evaluation, Fetal (blue), WBC (Orange), and gDNA (grey).
(TIF)

**S3 Fig. Evaluation of the performance of haplotyping at identifying a DNA with a non-maternal origin.** A. Performance of different haplotyping amplicons at detecting a non-maternal DNA. Y-axis is the detection rate, which estimates the fraction of the sample can be differentiated with a particular amplicon. The x-axis indicates which amplicon was tested. Fetal cells, WBC cells, and gDNA were tested accordingly from selected cases with both fetal cells and WBCs present. B. Improving detection rate by combining SNP typing and Haplotyping.

WBCs from different cases (with or without fetal cells) were analyzed with individual and combined approaches.
(TIF)

**S1 File. Reference sequence fasta file.** The fasta file used for alignment.
(FA)

**S2 File. SNVs reference BED.** The bed demonstrate position of SNVs.
(BED)

**S3 File. Linux shell script and parameters for alignment and analysis.**
(SH)

**S4 File. R script for genotype calling.**
(R)

**S5 File. R script for haplotype calling.**
(R)

**S6 File. R script to compare genotype of NIPT cells with parents.**
(R)

**S7 File. R script to compare haplotype of NIPT cells with parents.**
(R)

**S8 File. Table of selected clinical sample information.**
(XLSX)

## Acknowledgments

We thank the participating patients, study coordinators, and genetic counselors at Baylor College of Medicine, Texas Children's Hospital, and Columbia University Medical Center for recruitment of samples.

## Author Contributions

**Conceptualization:** Xinming Zhuo, Arthur Beaudet.

**Data curation:** Qun Wang, Liesbeth Vossaert, Roseen Salman.

**Formal analysis:** Xinming Zhuo, Qun Wang, Liesbeth Vossaert.

**Funding acquisition:** Ignatia Van den Veyver, Arthur Beaudet.

**Investigation:** Qun Wang, Roseen Salman, Adriel Kim.

**Methodology:** Xinming Zhuo, Qun Wang, Liesbeth Vossaert, Roseen Salman, Amy Breman.

**Project administration:** Arthur Beaudet.

**Resources:** Liesbeth Vossaert, Roseen Salman, Arthur Beaudet.

**Software:** Xinming Zhuo.

**Supervision:** Ignatia Van den Veyver, Amy Breman, Arthur Beaudet.

**Validation:** Xinming Zhuo, Qun Wang, Liesbeth Vossaert, Adriel Kim.

**Writing – original draft:** Xinming Zhuo.

**Writing – review & editing:** Xinming Zhuo, Qun Wang, Liesbeth Vossaert, Adriel Kim, Ignatia Van den Veyver, Amy Breman, Arthur Beaudet.

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
