## [Decision Letter · Decision Letter 0]

6 Aug 2020

PONE-D-20-14518

Use of amplicon-based sequencing for testing fetal identity and monogenic traits with single circulating trophoblast (SCT) prenatal diagnosis

PLOS ONE

Dear Dr. Zhuo,

Thank you for submitting your manuscript to PLOS ONE. After careful consideration, we feel that it has merit but does not fully meet PLOS ONE’s publication criteria as it currently stands. Therefore, we invite you to submit a revised version of the manuscript that addresses the points raised during the review process.

All reviewers expressed major concerns regarding the manuscript. It is required that you address all these concerns so that we may further consider this manuscript.

We look forward to receiving your revised manuscript.

Kind regards,

Osman El-Maarri, Ph.D

Academic Editor

PLOS ONE

Journal Requirements:

2. Please provide additional details regarding participant consent. In the ethics statement in the Methods and online submission information, please ensure that you have specified (1) whether consent was informed and (2) what type you obtained (for instance, written or verbal, and if verbal, how it was documented and witnessed). If the need for consent was waived by the ethics committee, please include this information.

3. We note that Figure(s) [1] in your submission contain copyrighted images. All PLOS content is published under the Creative Commons Attribution License (CC BY 4.0), which means that the manuscript, images, and Supporting Information files will be freely available online, and any third party is permitted to access, download, copy, distribute, and use these materials in any way, even commercially, with proper attribution. For more information, see our copyright guidelines: http://journals.plos.org/plosone/s/licenses-and-copyright.

a.    You may seek permission from the original copyright holder of Figure(s) [1] to publish the content specifically under the CC BY 4.0 license.

4. Thank you for including the following competing interests statement; "The Houston authors are faculty and staff at Baylor College of Medicine (BCM), which is a partial owner of a for profit diagnostic company, Baylor Genetics (BG); Houston authors also are employee of or have advisory or lab director roles at BG. ALB is founder and CEO of Luna Genetics, Inc."

Reviewers' comments:

Reviewer's Responses to Questions

**Comments to the Author**

1. Is the manuscript technically sound, and do the data support the conclusions?

Reviewer #1: Partly

Reviewer #2: Partly

Reviewer #3: No

2. Has the statistical analysis been performed appropriately and rigorously? 

Reviewer #1: Yes

Reviewer #2: Yes

Reviewer #3: I Don't Know

3. Have the authors made all data underlying the findings in their manuscript fully available?

Reviewer #1: Yes

Reviewer #2: No

Reviewer #3: Yes

4. Is the manuscript presented in an intelligible fashion and written in standard English?

Reviewer #1: Yes

Reviewer #2: Yes

Reviewer #3: Yes

5. Review Comments to the Author

Reviewer #1: Dear editor,

The paper “Use of amplicon-based sequencing for testing fetal identity and monogenic traits with

single circulating trophoblast (SCT) prenatal diagnosis” by Zhuo et al. is well written and clearly describes a promising procedure to distinguish fetal from maternal cells. However, I have a few remarks that require clarification.

Major point

- There is no discussion on the presence of fetal cells still present from previous pregnancies. Fetal cells have been detected decades after a pregnancy. Can the method described also distinguish if a fetal cell has a different haplotype than other fetal cells? If so, this is a strength of the methods described. If not, this would be a limitation.

- In addition, what would be the effect of twin pregnancies?

- Extra discussion is needed on the effects of the analysis on downstream analyses. Does the procedure cause lower quality results in downstream analysis of selected cells, since extra WGA amplification is needed to perform this test

Minor points,

- Line 999: Settings used for BWA-MEM. Does ‘conventional BWA-MEM’ mean all default settings?

- Table 1 is rather a list of amplicons containing SNP regions than a list of SNPs.

- In table 1: the start or stop of the HLA-A amplicon seems to be incorrects (Start>stop and not matching amplicon size).

- Lines 321 and 322, perhaps add the Protein accession numbers to the variants

- One method to infer haplotypes in cfDNA was not yet mentioned. https://pubmed.ncbi.nlm.nih.gov/28844486/

- Spelling mistake in fig 4: mother � mother and (informatic � informative?)

- The github code would benefit from some polishing, adding comments and removing commented out codes.

Reviewer #2: In this paper, the authors study the application of WGA followed by SNP and / or haplotype analysis in multiple amplicons to determine the fetal or maternal origin of presumably fetal cells, retrieved from the maternal blood. They do so by using different amplicons, i.e. amplicons from regions such as the HLA regions, with multiple SNPs per amplicon for haplotyping, and Human Identification SNP test amplicons, with one SNP per amplicon. They also studied the possibility of using these methods for genotyping for monogenic traits. Below my comments and questions:

1. My main comment is that, even though the authors do mention the drawbacks of their method, the paper is quite optimistic. The theory behind the methods they describe is solid, but the actual data are still quite preliminary. In their abstract the authors say that the method allowed reliable differentiation between fetal and maternal cells, in the introduction in line 75 they mention “in most cases”, but this was only in cases with sufficient information, which is certainly not 100%. Not all data is presented in a clear way, such as numbers of cells that are informative, but in how many cases? See my comments below. The methods that are described certainly have potential for this application, but need to be improved and the authors do acknowledge this in the discussion. In the discussion, the authors do mention multiple options for improvements, but how feasible are these? If feasible, why didn’t the authors already implement these, or at least show some data of improved versions, as proof-of-principle. The authors show data on fetal genotyping for monogenic traits, but in the discussion they mention themselves that a low failure rate is to be expected due to failure to isolate fetal cells and/or to ADO.

2. Throughout the paper, the authors use the term cell-based NIPT, whereas in the title they use the term single circulating trophoblast prenatal diagnosis. I would suggest to use the same terminology throughout the paper, and would suggest to also use the term cell-based NIPT in the title.

3. Line 40: analysis cell free = analysis of cell-free

4. In the introduction, the authors mention some drawbacks of cf-NIPT. One major drawback, of course, is the fact that cf-NIPT studies cd-DNA from the placenta and not from the fetus itself. This major drawback is not solved by the cell-based technique the authors use, as trophoblast cells are used for this method. The authors should comment on this. Furthermore, even though it is outside the scope of this paper, in the introduction the authors should also (briefly) comment on drawbacks of the cell-based NIPT, such as costs as compared to cf-NIPT and high-throughput possibilities.

5. With SNP typing, DNA from 156 blood samples were compared, with data from 518 cells: 357 cells were informative. Does being informative mean that the cells were fetal, or were there also maternal cells identified? If all fetal, were these cells originally also from 68.9% of the 156 samples, or were cells from some samples overrepresented and others underrepresented? In other words, could informative (fetal?) cells be retrieved from 68.9% of the 156 samples? (This refers to the comment that not all data support the conclusions, questions 1 and 3 of the review).

6. For haplotyping, how many cells from how many samples were tested? Again 156 samples? Also here, were the informative cells distributed evenly over the samples. In how many samples could fetal cells be identified based on haplotyping only? Is 50% of all cells the same as 50% of the cases? (Again, see questions 1 and 3 of the review)

7. The authors state they tested the haplotyping method with matched gDNA, WBC and fetal cells. In Line 278, they only mention WBC, not the fetal cells. What is the differentiating rate for the fetal cells? Please mention this in the text, not only in a supplementary figure, as this is the main subject of this paper. Moreover, according to Figure S3A, this is about 50% for both WBC and fetal cells, and this percentage is determined by one amplicon (HLA-B).

8. Figure S3A shows fetal cells with a detection rate of about 50%. In Figure S3B for HAP this seems to be more than 60%. What is the difference?

9. Lines 302-304: the authors state that 5 cells were genotyped: two were heterozygous and two had ADO. What happened to the fifth one?

10. In line 288, the authors state that in the cultured lymphoblasts, the ADO rate was 15% and 8%. When using true fetal cells from the three families, the ADO is very much higher, about 50%, as in each family some cells show ADO. How do the authors explain this difference?

11. From two of the three families more cells were retrieved than were used for genotyping. Why did the authors not genotype all available cells? Could not all cells be identified as true fetal? If so, that the identification of true fetal cells is lower than 60-70%. How were these cells identified as true fetal? If the other cells were maternal in origin, the chance of picking up a maternal cell is much higher than 10%, as stated in the introduction in line 65.

Reviewer #3: This paper describes the use of SNPs genotyping by sequencing in order to identifiy circulating cells of fetal origin in the context of NIPT applications.

Although the subject matter and the approach are interesting, there are significant issues with the way some of the results are presented and especially with the conclusions reached by the authors.

Because of the major limitations of the proposed methods, highlighted by several results (e.g. the combined performance of SNPs genotyping and haplotyping that gives only a 70% rate of success for identifying fetal cells from WBC (line 278, fig S3 B), a high rate of allele dropout and amplification artefacts contributing to false-negatives and false positives (line 335-341)), a clear and detailed presentation of false positive/negative rates should be provided.

Unfortunately, these rates are not comprehensively and clearly presented in the paper. Table 2 is not clearly explained. For example, I'm puzzled by the apparently negative correlation between true positives (TPV) and % of SNP differences (diff SNP%) in the "Fetal cell" column, shouldn't it be the opposite ? More info about the table headers, interpretation, etc. should be given. One of the main focus of the paper should be to present, analyse and discuss these rates in details.

This relatively poor performance of the genotyping approach to confirm fetal origin seems to be caused at least in part by the low amount/quality of starting DNA material since the detection rate for genomic DNA is much higher (Fig S3 A). This impact of starting material is also evident from the much lower number of scorable SNPs from WGA products as compared to genomic DNA shown on fig. 2.

These results thus point to serious technical challenges associated with WGA from single cells. Moreover, as the author acknowledge at line 335-341, both a high rate of allele dropout and amplification artefacts contribute to false-negatives and false positives.

The allele dropout problem mentionned above was also seriously affecting the genotyping for monogenic diseases, with rates reaching 8-15% (line 288). Such rate is clearly incompatible with any diagnostic application.

From these results, the realistic take home message is clearly one of skepticism towards the feasibility of using this kind of approach clinically in the short to mid term. However, although the authors acknowledge some of these limitations in the discussion, they nevertheless make statements that I consider misleading with respect to what the data show. For instance, "This single cell genotyping assay can provide an essential step for confirming the fetal origin of cells obtained from cell-based NIPT workflows. Through genotyping, we can reject cells that are of indeterminate or maternal origin and provide metrics for WGA DNA quality using multiple amplicons." (line 328-331). "...this assay is rapid and reliable..." (line 335). "Use of this method for genotyping fetuses at risk for specific monogenic mutations is feasible." (line 352-353).

More details should also be provided about the exact number of amplicons used rather than vague statements such as: “We sequenced approximately 90 highly polymorphic SNPs within about 40 amplicons.” (line 72). Likewise, in the methods section, three groups of amplicons are mentioned but only for the second group, a number is provided (37), while a reference is offered for the first group (Debeljak et al.), no number is given concerning the third group. Table 1 listing the amplicon used contains 49 entries.

Overall, the way the methods and results are presented lacks clarity and details that complicates the evaluation of the technique.

I thus consider that in its current form, this paper suffers from 1) a lack of clarity and a lack of emphasis on false positive/negative rates and other performance statistics, and 2) needs a more sober and realistic interpretation of the results, less focused on optimistic hopes of better future performances and more concerned about current challenges.

6. PLOS authors have the option to publish the peer review history of their article (what does this mean?). If published, this will include your full peer review and any attached files.

Reviewer #1: **Yes: **Lennart F. Johansson

Reviewer #2: No

Reviewer #3: No

---

## [Author Response · Author response to Decision Letter 0]

2 Nov 2020

To Editor’s comments:

We changed the format of manuscript and naming of files to meet PLOS One Style.

2. Please provide additional details regarding participant consent. In the ethics statement in the Methods and online submission information, please ensure that you have specified (1) whether consent was informed and (2) what type you obtained (for instance, written or verbal, and if verbal, how it was documented and witnessed). If the need for consent was waived by the ethics committee, please include this information.

We provide additional details about consent in materials and methods. “Blood samples were collected from pregnant women from multiple centers under a protocol approved by the Baylor College of Medicine Institutional Review Boards utilizing written informed consent.” It is now indicated that this was written informed consent.

3. We note that Figure(s) [1] in your submission contain copyrighted images. All PLOS content is published under the Creative Commons Attribution License (CC BY 4.0), which means that the manuscript, images, and Supporting Information files will be freely available online, and any third party is permitted to access, download, copy, distribute, and use these materials in any way, even commercially, with proper attribution. 

 We redraw Figure 1 to address the conflict.

4. Thank you for including the following competing interests statement; "The Houston authors are faculty and staff at Baylor College of Medicine (BCM), which is a partial owner of a for profit diagnostic company, Baylor Genetics (BG); Houston authors also are employee of or have advisory or lab director roles at BG. ALB is founder and CEO of Luna Genetics, Inc."

" The Houston authors are or were faculty and staff at Baylor College of Medicine(BCM), which is a partial owner of a for profit diagnostic company, Baylor Genetics (BG); Houston authors are also employees of or have advisory or lab director roles at BG. ALB is founder and CEO of Luna Genetics, Inc. This does not alter our adherence to PLOS ONE policies on sharing data and materials.”

 

To Reviewer #1’s comments:

The paper “Use of amplicon-based sequencing for testing fetal identity and monogenic traits with single circulating trophoblast (SCT) prenatal diagnosis” by Zhuo et al. is well written and clearly describes a promising procedure to distinguish fetal from maternal cells. However, I have a few remarks that require clarification.

Major point

- There is no discussion on the presence of fetal cells still present from previous pregnancies. Fetal cells have been detected decades after a pregnancy. Can the method described also distinguish if a fetal cell has a different haplotype than other fetal cells? If so, this is a strength of the methods described. If not, this would be a limitation.

- In addition, what would be the effect of twin pregnancies?

- Extra discussion is needed on the effects of the analysis on downstream analyses. Does the procedure cause lower quality results in downstream analysis of selected cells, since extra WGA amplification is needed to perform this test.

The following is added to the Introduction. This genotyping uses a small aliquot of the WGA product and does not interfere with downstream analysis. 

To the discussion. Although there is evidence that some cells from previous pregnancies can persist for decades, especialy CD34+ cells,{PMID: 8570620} there is no evidence that trophoblasts can persist from previous pregnancies, and we expect based on the biology of these cells including apoptosis, that they are unlikely to persist. A future method that provides deeper genotyping of individual cells could distinguish cells from previous pregnancies and cells from same sex nonidentical twins, but the method described here would not be sufficient to accomplish such distinction reliably.

Minor points,

- Line 999: Settings used for BWA-MEM. Does ‘conventional BWA-MEM’ mean all default settings?

Since it is amplicon seq and the data size relatively small, we can do the mapping on a regular laptop rather than a server, so we adjusted the BWA-MEM setting for small memory and minimum seed length (see the supplement script S3 for exact setting).

- Table 1 is rather a list of amplicons containing SNP regions than a list of SNPs.

The title of the table has been changed to List of Amplicons

- In table 1: the start or stop of the HLA-A amplicon seems to be incorrects (Start>stop and not matching amplicon size)

Here we used the 1-base format rather than 0-base (1-base format include the first and last base of the amplicon so that subtraction of one number from the other gives a number one less than the product size), so user can be easier to access the location in UCSC genome browser (1-base also). I add a footnote to the table to clarify the format.

- Lines 321 and 322, perhaps add the Protein accession numbers to the variants

We added ClinGen accession number to the variant in figure legend.

- One method to infer haplotypes in cfDNA was not yet mentioned. https://pubmed.ncbi.nlm.nih.gov/28844486/

We add this article to the reference:16. Vermeulen C, Geeven G, de Wit E, Verstegen MJAM, Jansen RPM, van Kranenburg M, et al. Sensitive Monogenic Noninvasive Prenatal Diagnosis by Targeted Haplotyping. Am J Hum Genet. 2017;101(3):326-39. Epub 2017/08/24. doi: 10.1016/j.ajhg.2017.07.012.

- Spelling mistake in fig 4: mother � mother and (informatic � informative?)

We fixed this typo.

- The github code would benefit from some polishing, adding comments and removing commented out codes. 

We have made revisions to address this.

 

To Reviewer #2’s comments: 

In this paper, the authors study the application of WGA followed by SNP and / or haplotype analysis in multiple amplicons to determine the fetal or maternal origin of presumably fetal cells, retrieved from the maternal blood. They do so by using different amplicons, i.e. amplicons from regions such as the HLA regions, with multiple SNPs per amplicon for haplotyping, and Human Identification SNP test amplicons, with one SNP per amplicon. They also studied the possibility of using these methods for genotyping for monogenic traits. Below my comments and questions:

1. My main comment is that, even though the authors do mention the drawbacks of their method, the paper is quite optimistic. The theory behind the methods they describe is solid, but the actual data are still quite preliminary. In their abstract the authors say that the method allowed reliable differentiation between fetal and maternal cells, in the introduction in line 75 they mention “in most cases”, but this was only in cases with sufficient information, which is certainly not 100%. Not all data is presented in a clear way, such as numbers of cells that are informative, but in how many cases? See my comments below. The methods that are described certainly have potential for this application, but need to be improved and the authors do acknowledge this in the discussion. In the discussion, the authors do mention multiple options for improvements, but how feasible are these? If feasible, why didn’t the authors already implement these, or at least show some data of improved versions, as proof-of-principle. The authors show data on fetal genotyping for monogenic traits, but in the discussion, they mention themselves that a low failure rate is to be expected due to failure to isolate fetal cells and/or to ADO.

The following sentences were added to the discussion, acknowledging the points made by the reviewer.

 “The data demonstrate the principle that detailed genotyping of individual cells can distinguish fetal from maternal cells, but the data are somewhat limited and preliminary, and new methods to more extensively compare the genotype of individual cells to the genotype of the mother are needed.”

The text has been modified as follows to address allele drop out and failure rate. 

“Single cell analysis of circulating trophoblasts can be used to determine the genotype of fetuses at risk for monogenic disorders, although allele dropout must be ruled out by analysis of multiple cells or it can be addressed using karyomapping as has been used in single cell preimplantation testing.{PMID: 19858130} Hopefully improved methods for recovering fetal cells from mother’s blood will reduce any failure due to lack of cells, but use of cell-free NIPT in combination with haplotype analysis or digital PCR, amniocentesis, and CVS would be three alternative strategies for difficult cases. “ 

2. Throughout the paper, the authors use the term cell-based NIPT, whereas in the title they use the term single circulating trophoblast prenatal diagnosis. I would suggest to use the same terminology throughout the paper, and would suggest to also use the term cell-based NIPT in the title.

Single circulating trophoblast testing is one form of cell based NIPT. They do not mean the same thing. We have modified the title to clarify this. Fetal nucleated RBC-based NIPT would be another form of cell based NIPT.

3. Line 40: analysis cell free = analysis of cell-free

Fixed

4. In the introduction, the authors mention some drawbacks of cf-NIPT. One major drawback, of course, is the fact that cf-NIPT studies cd-DNA from the placenta and not from the fetus itself. This major drawback is not solved by the cell-based technique the authors use, as trophoblast cells are used for this method. The authors should comment on this. Furthermore, even though it is outside the scope of this paper, in the introduction the authors should also (briefly) comment on drawbacks of the cell-based NIPT, such as costs as compared to cf-NIPT and high-throughput possibilities.

There are many complexities around many different forms of testing and mosaicism. The following sentence has been added to the introduction. “Cell-free NIPT, trophoblast-based NIPT, and CVS all can detect placental mosaicism, while amniocentesis, fetal nucleated red blood cell (fnRBC)-based NIPT when feasible, and fetal blood sampling can help to clarify whether mosaicism involves the fetus or is confined to the placenta.”

The following was also added. “Limitations of cost and throughput would need to be overcome for cell-based NIPT to be a routine alternative. At present to our knowledge, no laboratory offers cell-based NIPT as a clinical test.”

5. With SNP typing, DNA from 156 blood samples were compared, with data from 518 cells: 357 cells were informative. Does being informative mean that the cells were fetal, or were there also maternal cells identified? If all fetal, were these cells originally also from 68.9% of the 156 samples, or were cells from some samples overrepresented and others underrepresented? In other words, could informative (fetal?) cells be retrieved from 68.9% of the 156 samples? (This refers to the comment that not all data support the conclusions, questions 1 and 3 of the review).

The following sentence has been added as noted above in response to similar comments earlier. “The data demonstrate the principle that detailed genotyping of individual cells can distinguish fetal from maternal cells, but the data are somewhat limited, and new methods to more extensively compare the genotype of individual cells to the genotype of the mother are needed.”

The following has been added. “The results for cases can be divided into four groups as shown in Table 2. First, there were 7/154 (4.5%) cases where none the cells passing WGA could be proven to be fetal; Second, there were 42/154 (27.3%) cases with only one cell scored as fetal; and third, there were 103/154 (66.9%) with two or more cells scored as fetal as defined in methods. At the single cell level for cells that pass WGA, cells can be scored in only two ways: 1) uninformative meaning that no or inadequate evidence of fetal status or 2) adequate evidence for fetal status. Some of the “uninformative” cells were certainly maternal, and we cannot distinguish an uninformative fetal cell from a maternal cell. Deeper genotyping would make healthy uninformative fetal cells informative, but some fetal cells are known to be apoptotic with degraded DNA and may or may not pass WGA.” Case number reduce from original 156 to 154, due to 2 cases have poor quality maternal gDNA, which hinder the analysis.

6. For haplotyping, how many cells from how many samples were tested? Again 156 samples? Also here, were the informative cells distributed evenly over the samples. In how many samples could fetal cells be identified based on haplotyping only? Is 50% of all cells the same as 50% of the cases? (Again, see questions 1 and 3 of the review) 

Insertion of new Table 2 addresses this.

7. The authors state they tested the haplotyping method with matched gDNA, WBC and fetal cells. In Line 278, they only mention WBC, not the fetal cells. What is the differentiating rate for the fetal cells? Please mention this in the text, not only in a supplementary figure, as this is the main subject of this paper. Moreover, according to Figure S3A, this is about 50% for both WBC and fetal cells, and this percentage is determined by one amplicon (HLA-B).

Both the text and figure S2 and S3A mentioned fetal cells with WBC (Maternal cell). Since the Haplotyping only offer relative smaller contribution to the differentiation than SNP typing, we keep them in the supplement. 

The Figure S3B were used to estimate the power of a different genetic origin cell, WBC(single Maternal cell), from unmatched gDNA. Because the WBCs is picked intentionally and went through all the processing steps as potential fetal cells, which offer two benefits as control. First, their origin is very certain maternal origin. Second, they usually preserve a better shape, in contrast fetal cells occasionally were in the process of apoptosis or aggregated together. Comparing WBC with unmatched gDNA allow us to compare and estimate the limit of our method when encounter an intact cell. 

8. Figure S3A shows fetal cells with a detection rate of about 50%. In Figure S3B for HAP this seems to be more than 60%. What is the difference?

First, S3B for HAP is comparing the detection rate of WBC (~62%); while in S3A, WBCs combined is about 55%, they are about 7% difference. Second, S3A use only case with both Fetal cell and WBCs. S3B include cases with or without Fetal cells. Because they use different groups of cases for calculation, they cannot be compared directly. 

We edit the text as following:” when four amplicons were combined, we can still distinguish about 50% of all the cells (from selected cases with both fetal cells and WBCs) (S3 Fig A). With combined power of both SNP typing and haplotyping, we can increase the solving rate of differentiating a WBC from around 60% to more than 70% (from cases independent of fetal cell existence) (S3 Fig B).”

9. Lines 302-304: the authors state that 5 cells were genotyped: two were heterozygous and two had ADO. What happened to the fifth one?

The fifth cell does not carry mutation on that position. It is a possibility that the mutant allele drop out in this cell.

10. In line 288, the authors state that in the cultured lymphoblasts, the ADO rate was 15% and 8%. When using true fetal cells from the three families, the ADO is very much higher, about 50%, as in each family some cells show ADO. How do the authors explain this difference?

Sentence added. “The higher rate of allele drop out in fetal cells is presumably caused by some combination of DNA degradation caused by time in the maternal circulation including apoptosis and by cell isolation including fixation and permeabilization steps.”

11. From two of the three families more cells were retrieved than were used for genotyping. Why did the authors not genotype all available cells? Could not all cells be identified as true fetal? If so, that the identification of true fetal cells is lower than 60-70%. How were these cells identified as true fetal? If the other cells were maternal in origin, the chance of picking up a maternal cell is much higher than 10%, as stated in the introduction in line 65.

We do a quality check after WGA and determine if the cell should move on the next step. For cells pass the QC, we do the genotyping for most of them. Some cells either suffer bad quality (very low read number in sequencing) or have dropout reads in the position (such as cell dicussed in question #9). 

 

To Reviewer #3’s comments: 

This paper describes the use of SNPs genotyping by sequencing in order to identifiy circulating cells of fetal origin in the context of NIPT applications.

Although the subject matter and the approach are interesting, there are significant issues with the way some of the results are presented and especially with the conclusions reached by the authors.

Because of the major limitations of the proposed methods, highlighted by several results (e.g. the combined performance of SNPs genotyping and haplotyping that gives only a 70% rate of success for identifying fetal cells from WBC (line 278, fig S3 B), a high rate of allele dropout and amplification artefacts contributing to false-negatives and false positives (line 335-341)), a clear and detailed presentation of false positive/negative rates should be provided.

Based on this comment and similar comments above, we made a new table2. Significant modifications have been introduced into the Results and Discussion as below. 

“The results for cases can be divided into four groups as shown in Table 2. First, there were 7/154 (4.5%) cases where none the cells passing WGA could be proven to be fetal; Second, there were 42/154 (27.3%) cases with only one cell scored as fetal; and third, there were 103/154 (66.9%) with two or more cells scored as fetal as defined in methods. At the single cell level for cells that pass WGA, cells can be scored in only two ways: 1) uninformative meaning that no or inadequate evidence of fetal status or 2) adequate evidence for fetal status. Some of the “uninformative” cells were certainly maternal, and we cannot distinguish an uninformative fetal cell from a maternal cell. Deeper genotyping would make healthy uninformative fetal cells informative, but some fetal cells are known to be apoptotic with degraded DNA and may or may not pass WGA. False positives and false negatives at a single cell level are discussed below. This method is a work in progress, and we consider one fetal cell as partial success and two or more fetal cells as success from a clinical perspective. Improved cell recovery and improved genotyping would be needed to achieve an optimal test. The relative roles of failure to isolate and amplify cells, uninformative genotyping, and suboptimal numbers of fetal cells can be calculated from Table 2.”

Unfortunately, these rates are not comprehensively and clearly presented in the paper. Table 2 is not clearly explained. For example, I'm puzzled by the apparently negative correlation between true positives (TPV) and % of SNP differences (diff SNP%) in the "Fetal cell" column, shouldn't it be the opposite ? More info about the table headers, interpretation, etc. should be given. One of the main focus of the paper should be to present, analyse and discuss these rates in details.

We agree that the rates were no presented clearly, and we have constructed a different presentation with a new table 3, which we believe is much improved.

The following is inserted in the results.

“If one assumes that the NGS data are 100% reliable for sex based on X and Y data, which we believe is the case, the sensitivity and specificity of the genotyping can be estimated from normal male singleton pregnancies as shown in Table 3. Definition of other contingencies are included in the same table. Identifying a fetal cell as fetal by genotyping and confirming that it is male based on NGS is a true positive. The results vary depending on the cutoff for scoring a SNP allele as present. If we accept two different 2 SNP + 6% cutoff, based on normal male singleton pregnancies the true positive rate was 82.1%, true negative 6.6%, false positive 3.3%, and false negative 7.9%. This is equal to a sensitivity of 91.2% and a precision of 96.1%. These results would not be sufficient for clinical diagnosis, but they demonstrate the general validity of the approach and suggest that deeper genotyping of single cell could be completely reliable.”

This relatively poor performance of the genotyping approach to confirm fetal origin seems to be caused at least in part by the low amount/quality of starting DNA material since the detection rate for genomic DNA is much higher (Fig S3 A). This impact of starting material is also evident from the much lower number of scorable SNPs from WGA products as compared to genomic DNA shown on fig. 2.

The reviewer is correct. The poor performance on genotyping is caused in part by low amount/quality of DNA.

The following sentence has been added “The higher rate of allele drop out in fetal cells is presumably caused by some combination of DNA degradation caused by time in the maternal circulation including apoptosis and by cell isolation including fixation and permeabilization steps.”

These results thus point to serious technical challenges associated with WGA from single cells. Moreover, as the author acknowledge at line 335-341, both a high rate of allele dropout and amplification artefacts contribute to false-negatives and false positives.

One can address false positive and false negatives theoretically in genotyping a single cell. False positives where a sequencing error introduces the alternative of two alleles at a position would be very rare where alternate allele appears by chance. Single base sequencing errors are moderately common but do not create a false positive. False negative could occur due to allele drop out which is higher in cells recovered from mother’s blood than for lymphoblasts. Clearly ADO is a very common occurrence. 

Limitations are failure to isolate fetal cells, failure of WGA (e.g., due to apoptosis), allele drop out, and insufficient depth of genotyping. We hypothesize that all of these can potentially be overcome by recovering more cells and by deeper genotyping of each cell (i.e. genotyping thousands of SNPs).

The allele dropout problem mentionned above was also seriously affecting the genotyping for monogenic diseases, with rates reaching 8-15% (line 288). Such rate is clearly incompatible with any diagnostic application.

ADO is a problem for diagnosing monogenic disorders using cell-based NIPT. If a nucleotide position is heterozygous in even one cell, but preferably in multiple cells, the genotype at that position can be scored with confidence. If the genotype shows only the normal (N) allele at a position, interpretation can conclude that the genotype is NN or NM but not MM (affected). If the genotype shows only the mutant (M) at a position, interpretation can conclude that the genotype is MM of NM but not NN. This circumstance occurs for common variants causing phenotypes as for sickle cell anemia or cystic fibrosis. If one allele is found in some cells and the opposite allele is found in other cells in a singleton pregnancy, we would argue that finding two cells with only the N allele and two cells with only the M allele would allow the conclusion that the genotype is heterozygous at that position. For compound heterozygous genotypes, the circumstances are somewhat different. Observing both alleles in multiple cells is straight forward for interpretation that the fetus carries the mutant allele at this position. Observing only the M allele would also allow the conclusion that the fetus carries the mutant allele at this position Observing only the N allele in even many cells is challenging, since the fetus could carry the mutation with allele dropout in all five cells. This could occur especially if a SNP impairs the function of a primer. This concern can be reduced by demonstrating that the primers used amplify all parental alleles. The uncertainty of a genotype position could be addressed by cell-free NIPT for a paternal mutation, but a method such as haplotyping, karyomapping, digital PCR, or amniocentesis or CVS could be used for maternal mutations.

From these results, the realistic take home message is clearly one of skepticism towards the feasibility of using this kind of approach clinically in the short to mid term. However, although the authors acknowledge some of these limitations in the discussion, they nevertheless make statements that I consider misleading with respect to what the data show. For instance, "This single cell genotyping assay can provide an essential step for confirming the fetal origin of cells obtained from cell-based NIPT workflows. Through genotyping, we can reject cells that are of indeterminate or maternal origin and provide metrics for WGA DNA quality using multiple amplicons." (line 328-331). "...this assay is rapid and reliable..." (line 335). "Use of this method for genotyping fetuses at risk for specific monogenic mutations is feasible." (line 352-353).

We agree with skepticism about using the current method clinically, but it does demonstrate that a deeper genotyping method that detects thousands of SNPs in single cells could be used clinically.

More details should also be provided about the exact number of amplicons used rather than vague statements such as: “We sequenced approximately 90 highly polymorphic SNPs within about 40 amplicons.” (line 72). Likewise, in the methods section, three groups of amplicons are mentioned but only for the second group, a number is provided (37), while a reference is offered for the first group (Debeljak et al.), no number is given concerning the third group. Table 1 listing the amplicon used contains 49 entries.

The total number of amplicons are case specific. First 41 amplicons are regular for identifying the origin of cell. Then additional amplicons are added if the parents have monogenic disorder, so the total number is case by case.

Overall, the way the methods and results are presented lacks clarity and details that complicates the evaluation of the technique.

I thus consider that in its current form, this paper suffers from 1) a lack of clarity and a lack of emphasis on false positive/negative rates and other performance statistics, and 2) needs a more sober and realistic interpretation of the results, less focused on optimistic hopes of better future performances and more concerned about current challenges.

We think that the revisions above address these weaknesses.

---

## [Decision Letter · Decision Letter 1]

4 Jan 2021

PONE-D-20-14518R1

Use of amplicon-based sequencing for testing fetal identity and monogenic traits with single circulating trophoblast (SCT) as one form of cell-based NIPT

PLOS ONE

Dear Dr. Zhuo,

Thank you for submitting your manuscript to PLOS ONE. After careful consideration, we feel that it has merit but does not fully meet PLOS ONE’s publication criteria as it currently stands. Therefore, we invite you to submit a revised version of the manuscript that addresses the points raised during the review process.

We look forward to receiving your revised manuscript.

Kind regards,

Osman El-Maarri, Ph.D

Academic Editor

PLOS ONE

Reviewers' comments:

Reviewer's Responses to Questions

**Comments to the Author**

1. If the authors have adequately addressed your comments raised in a previous round of review and you feel that this manuscript is now acceptable for publication, you may indicate that here to bypass the “Comments to the Author” section, enter your conflict of interest statement in the “Confidential to Editor” section, and submit your "Accept" recommendation.

Reviewer #2: (No Response)

Reviewer #3: All comments have been addressed

2. Is the manuscript technically sound, and do the data support the conclusions?

Reviewer #2: Partly

Reviewer #3: Partly

3. Has the statistical analysis been performed appropriately and rigorously? 

Reviewer #2: Yes

Reviewer #3: Yes

4. Have the authors made all data underlying the findings in their manuscript fully available?

Reviewer #2: Yes

Reviewer #3: Yes

5. Is the manuscript presented in an intelligible fashion and written in standard English?

Reviewer #2: Yes

Reviewer #3: Yes

6. Review Comments to the Author

Reviewer #2: The authors have done a significant amount of work from which the paper certainly benefits.

Previously, I commented on the difference between cells and cases. Even though the data in the paper are presented in a much better way now, I still have some questions:

1. In the rebuttal the authors mention the number of cases is now 154 instead of 156. In the text, they use the number of 154, while in table 2 the numbers do not add up to 154 but to 152. Please explain or correct.

2. In Table 2, numbers of cases are presented, which in my opinion is indeed the most informative way of (at least) presenting the data. In lines 213-218 the authors again switch to information on number of cells instead of cases. This apparently is not the same as cases, as when calculating % from the data in Table 2, different % are obtained than mentioned in the text. Please use the same way of presenting data in both text and Table.

3. In Table 2 the last column seems to be the same as the “2 SNP + 6%” column as far as data is concerned. What does this last column show? According to Table 3 it should be “2 SNP + 0%”, but what does that mean? The numbers in the column in Table 2 are the same as in the “2 SNP + 6%” column. Is this correct?

4. Table 2 would benefit from mentioning also the percentages, for better comparison with the text.

5. Lines 226-230 and Table 3: this is again based on cells. Please also provide information of cases.

6. Lines 337-342: As mentioned in my previous comment, the data n the numbers of cells do not add up to the numbers of cells that are genotyped. Please add information on all the cells, for both the DHCR7 and the sickle cell family.

7. Line 337: “…but the data are somewhat limited…”. Please remove the word “somewhat”, as this single word raises a lot of questions and removing the word does not change the message of the sentence.

8. Lines 403-405: the authors suggest as alternatives to overcome the problem with their method to use their NIPT in combination with amniocenteses and CVS. This seems a strange alternative, as cell-based NIPT is being validated and optimized as replacement for invasive methods. What would be the benefit of cell-based NIPT if it should be used in combination with an invasive procedure anyway?

Minor comments:

9. Abstract line 26: “This method allowed reliable differentiation of fetal and maternal cells.” This is still an optimistic, and possibily misleading, sentence. The sentence can be removed without any damage to the abstract.

10. Throughout the paper, the term “mutation” is used. There is an international agreement to use the word “variant” or “pathogenic variant” instead (see Richards et al. 2015). Please use this throughout the paper.

11. Line 199: “…where none the cells…” = “…where none of the cells…”

12. Line 202: “…meaning that no…” = “…meaning no….”

13. Line 214: “…used a more conservative….” = “…used a more conservative approach of…”

14. Line 226: “…two different 2 SNP…” = “two different SNPs…”

15. Line 241: “…per sample…” = “…per cell…”

16. Line 351: “…the sickle mutation…” = “…the sickle cell anemia variant…”

17. Line 377: spelling error in pregnancies and piology (=etiology??)

Reviewer #3: Although the authors have attempted to address all the comments raised in the previous review, a few issues persist.

Notably, in my opinion, the discussion still contains overly optimistic statements that do not adequately portrait what the current data suggest. For instance, at line 381, I would replace: "...this assay is rapid and reliable…" by "...this assay is promising…".

at line 393, replace: "...monogenic mutations is feasible." by "...monogenic mutations is potentially feasible.". At line 410, replace: "An improved version of this assay may significantly reduce…" by "A future improved version may hopefully reduce…".

The paper would also greatly benefit from clarifying the terminology used when referring to "informative" cells and SNPs throughout. It seems like the authors refer to SNPs with more than one allele present in the NGS data from a sample as an informative SNP, whether or not there's a difference between the mother and the fetus at this SNP position (e.g. when both mother and fetus are heterozygotes for the same alleles), while an informative cell refers to a cell with feto-maternal SNP differences. These terms should be formally defined early on to avoid confusion. However, at line 299, the authors refer to informative SNPs as those where there's a difference between the genotype of the mother and of the putative fetal cell. The term scorable SNP is also used, which here seems to simply refer to a genotyped SNP position that passed quality control.

Additional minor suggestions and comments

line 98: "The first step is conventional PCR with a pool of amplicons with bridging adaptors…" should be : "The first step is conventional PCR of a pool of amplicons with bridging adaptors…"

line 140: "An 8-10 pM diluted denatured…"

line 188: "...three SNPs show a difference between one of the cell…"

line 198: I believe it should be three groups and not four: "The results for cases can be divided into three groups…"

line 214: "...we initially used a more conservative criterion of at least…"

line 215: "We found that 68.9 % of genotyped putative fetal cells…"

line 228: what does precision of 96.1 % refers to here ? To the authors mean global diagnostic accuracy ? Please mathematically define the precision calculation used.

line 235: "The remaining three cells…"

line 348: The last sentence ("The fetus does carry the paternal mutation.") seems odd and should be removed or modified.

7. PLOS authors have the option to publish the peer review history of their article (what does this mean?). If published, this will include your full peer review and any attached files.

Reviewer #2: No

Reviewer #3: No

---

## [Author Response · Author response to Decision Letter 1]

26 Jan 2021

Reviewer #2: The authors have done a significant amount of work from which the paper certainly benefits.

Previously, I commented on the difference between cells and cases. Even though the data in the paper are presented in a much better way now, I still have some questions:

1. In the rebuttal the authors mention the number of cases is now 154 instead of 156. In the text, they use the number of 154, while in table 2 the numbers do not add up to 154 but to 152. Please explain or correct.

The following sentence is added to the first paragraph of methods describing the samples. 

“There were 154 usable blood samples from 152 pregnancies/cases; two women had two samples collected during one pregnancy.”

To avoid confusion, we also revised the number accordingly in the text. 

2. In Table 2, numbers of cases are presented, which in my opinion is indeed the most informative way of (at least) presenting the data. In lines 213-218 the authors again switch to information on number of cells instead of cases. This apparently is not the same as cases, as when calculating % from the data in Table 2, different % are obtained than mentioned in the text. Please use the same way of presenting data in both text and Table.

The paragraph starting at line 213 is substantially rewritten as follows: 

In Table 2, we examined what percent of cases had one or more or two of more cells scored as fetal. Individual cells were scored as fetal if two or more SNPs had at least 10% reads for an allele that was not present in the mother. Cases were then subdivided into those where the informative SNPs indicating fetal status were at least 10%, 8%, or 6%, of the scorable SNPs. There were no cases where the informative SNPs were less than 6% of the scorable SNPs. There were 60 cells with one informative SNP suggesting that requiring two SNPs may undercount fetal cells. It is important to distinguish the percent of cases (82.4%) that had two or more fetal cells (103/152) from the percent of cells (78.7%) that had two or more fetal SNPs indicating fetal status (408/518).

3. In Table 2 the last column seems to be the same as the “2 SNP + 6%” column as far as data is concerned. What does this last column show? According to Table 3 it should be “2 SNP + 0%”, but what does that mean? The numbers in the column in Table 2 are the same as in the “2 SNP + 6%” column. Is this correct?

The last column of Table 2 has been deleted as not contributing useful information.

4. Table 2 would benefit from mentioning also the percentages, for better comparison with the text.

We edited the table as suggested.

5. Lines 226-230 and Table 3: this is again based on cells. Please also provide information of cases.

This Table presents the reliability of scoring a cell as fetal, and it is essential to present the data as cells. There are multiple cells per case, and it is not the case that is a false positive or false negative. No change is made to the text. 

6. Lines 337-342: As mentioned in my previous comment, the data n the numbers of cells do not add up to the numbers of cells that are genotyped. Please add information on all the cells, for both the DHCR7 and the sickle cell family.

For the sickle cell case, we made an error. Nine fetal cells were recovered and four were genotyped. The fifth cell was a maternal WBC intentionally picked as a control. The text is corrected. Information for all of the cells genotyped is now presented for the DHCR7 and sickle cell families. 

7. Line 337: “…but the data are somewhat limited…”. Please remove the word “somewhat”, as this single word raises a lot of questions and removing the word does not change the message of the sentence.

The word was removed.

8. Lines 403-405: the authors suggest as alternatives to overcome the problem with their method to use their NIPT in combination with amniocenteses and CVS. This seems a strange alternative, as cell-based NIPT is being validated and optimized as replacement for invasive methods. What would be the benefit of cell-based NIPT if it should be used in combination with an invasive procedure anyway?

The intent was to mention amniocentesis and CVS as alternatives if cell-based NIPT fails. The text is changed to indicate this. 

Minor comments:

9. Abstract line 26: “This method allowed reliable differentiation of fetal and maternal cells.” This is still an optimistic, and possibily misleading, sentence. The sentence can be removed without any damage to the abstract.

Removed.

10. Throughout the paper, the term “mutation” is used. There is an international agreement to use the word “variant” or “pathogenic variant” instead (see Richards et al. 2015). Please use this throughout the paper.

There were 43 occurrences of the word mutation and 8 occurrences of the word mutant. Dozens of changes were made where appropriate.

11. Line 199: “…where none the cells…” = “…where none of the cells…”

Changed

12. Line 202: “…meaning that no…” = “…meaning no….”

changed

13. Line 214: “…used a more conservative….” = “…used a more conservative approach of…”

The entire paragraph was rewritten. 

14. Line 226: “…two different 2 SNP…” = “two different SNPs…”

Changed

15. Line 241: “…per sample…” = “…per cell…”

changed

16. Line 351: “…the sickle mutation…” = “…the sickle cell anemia variant…”

Changed

17. Line 377: spelling error in pregnancies and piology (=etiology??)

Changed to biology

Reviewer #3: Although the authors have attempted to address all the comments raised in the previous review, a few issues persist.

Notably, in my opinion, the discussion still contains overly optimistic statements that do not adequately portrait what the current data suggest. For instance, at line 381, I would replace: "...this assay is rapid and reliable…" by "...this assay is promising…".

at line 393, replace: "...monogenic mutations is feasible." by "...monogenic mutations is potentially feasible.". At line 410, replace: "An improved version of this assay may significantly reduce…" by "A future improved version may hopefully reduce…".

Changed as suggested.

The paper would also greatly benefit from clarifying the terminology used when referring to "informative" cells and SNPs throughout. It seems like the authors refer to SNPs with more than one allele present in the NGS data from a sample as an informative SNP, whether or not there's a difference between the mother and the fetus at this SNP position (e.g. when both mother and fetus are heterozygotes for the same alleles), while an informative cell refers to a cell with feto-maternal SNP differences. These terms should be formally defined early on to avoid confusion. However, at line 299, the authors refer to informative SNPs as those where there's a difference between the genotype of the mother and of the putative fetal cell. The term scorable SNP is also used, which here seems to simply refer to a genotyped SNP position that passed quality control.

The following sentence is added in the methods section under variant calling. 

Throughout this manuscript, a cell or a SNP is referred to as informative if the putative fetal cell has an allele not carried by the mother (e.g., mother is AA and the putative fetal cell is AB or _B). An uninformative cell does not have alleles not carried by the mother and may be a maternal cell of a fetal cell with inadequate genotyping data.

The legend for Fig. 6 is very consistent with this explanation. 

Additional minor suggestions and comments

line 98: "The first step is conventional PCR with a pool of amplicons with bridging adaptors…" should be : "The first step is conventional PCR of a pool of amplicons with bridging adaptors…"

Changed

line 140: "An 8-10 pM diluted denatured…"

changed

line 188: "...three SNPs show a difference between one of the cell…"

changed

line 198: I believe it should be three groups and not four: "The results for cases can be divided into three groups…"

changed

line 214: "...we initially used a more conservative criterion of at least…"

This paragraph is rewritten

line 215: "We found that 68.9 % of genotyped putative fetal cells…"

This paragraph is rewritten

line 228: what does precision of 96.1 % refers to here ? To the authors mean global diagnostic accuracy ? Please mathematically define the precision calculation used.

Changed the text:

 “This is equal to a sensitivity [True Positive Rate= TP/(TP+FN) of 91.2% (124/124+12)] and a precision [Positive Predictive Value = TP/(TP+FP) of 96.1% (124/124+5)].”

line 235: "The remaining three cells…"

Changed

line 348: The last sentence ("The fetus does carry the paternal mutation.") seems odd and should be removed or modified.

Deleted sentence.

---

## [Decision Letter · Decision Letter 2]

9 Mar 2021

PONE-D-20-14518R2

Use of amplicon-based sequencing for testing fetal identity and monogenic traits with single circulating trophoblast (SCT) as one form of cell-based NIPT

PLOS ONE

Dear Dr. Zhuo,

Thank you for submitting your manuscript to PLOS ONE. After careful consideration, we feel that it has merit but does not fully meet PLOS ONE’s publication criteria as it currently stands. Therefore, we invite you to submit a revised version of the manuscript that addresses the points raised during the review process.

We look forward to receiving your revised manuscript.

Kind regards,

Osman El-Maarri, Ph.D

Academic Editor

PLOS ONE

Journal Requirements:

Reviewers' comments:

Reviewer's Responses to Questions

**Comments to the Author**

1. If the authors have adequately addressed your comments raised in a previous round of review and you feel that this manuscript is now acceptable for publication, you may indicate that here to bypass the “Comments to the Author” section, enter your conflict of interest statement in the “Confidential to Editor” section, and submit your "Accept" recommendation.

Reviewer #2: All comments have been addressed

Reviewer #3: (No Response)

2. Is the manuscript technically sound, and do the data support the conclusions?

Reviewer #2: Yes

Reviewer #3: Yes

3. Has the statistical analysis been performed appropriately and rigorously? 

Reviewer #2: Yes

Reviewer #3: Yes

4. Have the authors made all data underlying the findings in their manuscript fully available?

Reviewer #2: Yes

Reviewer #3: Yes

5. Is the manuscript presented in an intelligible fashion and written in standard English?

Reviewer #2: Yes

Reviewer #3: Yes

6. Review Comments to the Author

Reviewer #2: (No Response)

Reviewer #3: The authors have addressed most of the issues previously identified, but there remains a few details that need to be clarified.

As requested, the authors have added a sentence defining and clarifying the "informative/uninformative" terminology in the methods section, where this definition is given: "Throughout this manuscript, a cell or a SNP is referred to as informative if the putative fetal cell has an allele not carried by the mother...".

However, at line 249, while refering to Fig. 4, they state that "Cell G106 has only 2 SNP difference in less than 20 informative SNPs...". The axis on Fig. 4 are indeed labelled "number of SNPs not present in mother" and "number of informative SNPs", for the y and x axis respectively.

Therefore, given the definition of informative SNP provided by the authors, I don't understand what can be the difference here between "number of SNPs not present in mother" and "number of informative SNPs". Aren't SNPs not present in mother suppose to be alleles not present in mother ? Please clarify.

Regarding Fig. 4, there are other inconsistencies between the text and the figure. At line 245, it is said that cells G78, G212, G227 and G232 all have at least four SNPs wich are different from maternal gDNA while on the graph, it seems to be at least five SNPs which are different (of these four cells, G78 and G212 have the lowest number of differences and it seems to be five differences when judging from the axis and the points on the graph). Then it is said that the remaining three cells had 0-2 SNPs which differed from maternal gDNA, while it seems to be rather 0-1 judging from the figure ?

Moreover, the caption for Fig. 4 says: "As expected, the maternal gDNA samples gave no alleles not present in the mother...". This sentence seems odd given that thoughout the paper, we get the impression that maternal gDNA (from WBC) is used as the reference for the mother's genome. How could there be an allele not present in the mother from the data used to establish the mother's genome ? That would be circular ? Was there another reference used ?

Finally, at line 100, the requested change of "...conventional PCR with a pool..." for "...conventional PCR of a pool..." has not been made, even if the authors claim to have made the change in their reply.

7. PLOS authors have the option to publish the peer review history of their article (what does this mean?). If published, this will include your full peer review and any attached files.

Reviewer #2: No

Reviewer #3: No

---

## [Author Response · Author response to Decision Letter 2]

20 Mar 2021

Dear Editor,

We would like to thank the reviewers for their specific and helpful comments. The manuscript has been improved according to the suggestions from the reviewers. 

For specific comments, please see our response below:

Reviewer #3: The authors have addressed most of the issues previously identified, but there remains a few details that need to be clarified.

As requested, the authors have added a sentence defining and clarifying the "informative/uninformative" terminology in the methods section, where this definition is given: "Throughout this manuscript, a cell or a SNP is referred to as informative if the putative fetal cell has an allele not carried by the mother...".

However, at line 249, while refering to Fig. 4, they state that "Cell G106 has only 2 SNP difference in less than 20 informative SNPs...". The axis on Fig. 4 are indeed labelled "number of SNPs not present in mother" and "number of informative SNPs", for the y and x axis respectively.

Therefore, given the definition of informative SNP provided by the authors, I don't understand what can be the difference here between "number of SNPs not present in mother" and "number of informative SNPs". Aren't SNPs not present in mother suppose to be alleles not present in mother ? Please clarify.

We changed the labels in figure 4 and the text accordingly.

“The X-axis indicates the number of variant sites passing coverage cutoff. The Y-axis is the number of informative SNPs in a cell.”

Regarding Fig. 4, there are other inconsistencies between the text and the figure. At line 245, it is said that cells G78, G212, G227 and G232 all have at least four SNPs wich are different from maternal gDNA while on the graph, it seems to be at least five SNPs which are different (of these four cells, G78 and G212 have the lowest number of differences and it seems to be five differences when judging from the axis and the points on the graph). Then it is said that the remaining three cells had 0-2 SNPs which differed from maternal gDNA, while it seems to be rather 0-1 judging from the figure ?

Moreover, the caption for Fig. 4 says: "As expected, the maternal gDNA samples gave no alleles not present in the mother...". This sentence seems odd given that thoughout the paper, we get the impression that maternal gDNA (from WBC) is used as the reference for the mother's genome. How could there be an allele not present in the mother from the data used to establish the mother's genome ? That would be circular ? Was there another reference used ?

We changed typo in the text to “at least five” and “0-1” according to suggest. We also removed the sentence "As expected, the maternal gDNA samples gave no alleles not present in the mother..." to avoid confusion.

Finally, at line 100, the requested change of "...conventional PCR with a pool..." for "...conventional PCR of a pool..." has not been made, even if the authors claim to have made the change in their reply.

We fixed this typo.

We think that the revisions above address these weaknesses.

We hope that our revision in its current form is suitable for publication in PLOS ONE.

Sincerely yours, 

On behalf of the authors

Xinming Zhuo, Ph.D. and Arthur Beaudet, M.D.

---

## [Editor Report · Decision Letter 3]

24 Mar 2021

Use of amplicon-based sequencing for testing fetal identity and monogenic traits with single circulating trophoblast (SCT) as one form of cell-based NIPT

PONE-D-20-14518R3

Dear Dr. Zhuo,

We’re pleased to inform you that your manuscript has been judged scientifically suitable for publication and will be formally accepted for publication once it meets all outstanding technical requirements.

Kind regards,

Osman El-Maarri, Ph.D

Academic Editor

PLOS ONE

---

## [Editor Report · Acceptance letter]

6 Apr 2021

PONE-D-20-14518R3 

Use of amplicon-based sequencing for testing fetal identity and monogenic traits with single circulating trophoblast (SCT) as one form of cell-based NIPT 

Dear Dr. Zhuo:

I'm pleased to inform you that your manuscript has been deemed suitable for publication in PLOS ONE. Congratulations! Your manuscript is now with our production department. 

Kind regards, 

on behalf of

Priv.-Doz. Dr. Osman El-Maarri 

Academic Editor

PLOS ONE